**Data Availability Statement:** All relevant data are within the manuscript and its Supporting Information files.

# A systematic review on improving implementation of the revitalised integrated disease surveillance and response system in the African region: A health workers' perspective

**Arthur K. S. Ng'etich**[1]*, **Kuku Voyi**[1], **Ruth C. Kirinyet**[2], **Clifford M. Mutero**[1,3,4]

1 School of Health Systems and Public Health (SHSPH), University of Pretoria, Pretoria, South Africa, 2 Department of Environmental and Occupational Health, School of Public Health, Kenyatta University, Nairobi, Kenya, 3 University of Pretoria Institute for Sustainable Malaria Control (UP ISMC), University of Pretoria, Pretoria, South Africa, 4 International Centre of Insect Physiology and Ecology, Nairobi, Kenya

* arthursaitabau@yahoo.com

## Abstract

### Background

The revised integrated disease surveillance and response (IDSR) guidelines adopted by African member states in 2010 aimed at strengthening surveillance systems critical capacities. Milestones achieved through IDSR strategy implementation prior to adopting the revised guidelines are well documented; however, there is a dearth of knowledge on the progress made post-adoption. This study aimed to review key recommendations resulting from surveillance assessment studies to improve implementation of the revitalised IDSR system in the African region based on health workers' perspectives. The review focused on literature published between 2010 and 2019 post-adopting the revised IDSR guidelines in the African region.

### Methods

A systematic literature search in PubMed, Web of Science and Cumulative Index for Nursing and Allied Health Literature was conducted. In addition, manual reference searches and grey literature searches using World Health Organisation Library and Information Networks for Knowledge databases were undertaken. The Preferred Reporting Items for Systematic Reviews and Meta-Analyses statement checklist for systematic reviews was utilised for the review process.

### Results

Thirty assessment studies met the inclusion criteria. IDSR implementation under the revised guidelines could be improved considerably bearing in mind critical findings and recommendations emanating from the reviewed surveillance assessment studies. Key recommendations alluded to provision of laboratory facilities and improved specimen handling, provision

**Funding:** The author(s) received no specific funding for this work.

**Competing interests:** The authors have declared that no competing interests exist.

of reporting forms and improved reporting quality, surveillance data accuracy and quality, improved knowledge and surveillance system performance, utilisation of up-to-date information and surveillance system strengthening, provision of resources, enhanced reporting timeliness and completeness, adopting alternative surveillance strategies and conducting further research to improve surveillance functions.

## Conclusion

Recommendations on strengthening IDSR implementation in the African region post-adopting the revised guidelines mainly identify surveillance functions focused on reporting, feedback, training, supervision, timeliness and completeness of the surveillance system as aspects requiring policy refinement.

## Systematic review registration

PROSPERO registration number CRD42019124108.

## Introduction

Public health surveillance involves continuous collection, analysis and interpretation of health data resulting in timely information dissemination enabling effective public health action [1]. Public health surveillance systems form a critical part of information systems as a key component within the World Health Organization (WHO) health system framework [2]. The International Health Regulations (IHR 2005) within the health system are a legally binding agreement providing a framework to coordinate and manage public health threats [3,4]. The IHR (2005) necessitated all WHO member states to evaluate ability of their national structures, capacities and resources to achieve effective disease surveillance and response [3]. Prior to IHR (2005), the WHO Regional Office for Africa (WHO-AFRO) and its member states adopted the Integrated Disease Surveillance and Response (IDSR) system [5]. IDSR system framework provided a platform to improve national public health surveillance and response capacities. The IDSR system aims to strengthen the public health system at community, health facility, district, and national levels to ensure timely detection, confirmation and response to public health threats to alleviate illness, disability and mortality [5,6]. IDSR and IHR frameworks share a common goal of improving timely response to public health events through early detection, notification, verification, response and collaboration activities [3,6]. Therefore, member states in WHO African region declared IHR (2005) implementation was to be achieved within the existing IDSR strategy [6]. A review of IDSR guidelines in 2010 was necessary to meet the requirements of disease surveillance and response core capacities strengthening as specified through IHR (2005) implementation by African member states [5–7].

The revised IDSR guidelines considered the recommended tools and approaches from IHR (2005) to supplement early warning capabilities in the national surveillance systems and tackle other threats to public health [6]. By 2016, 42 out of 47 countries in the African region had adopted the second edition of the revitalised IDSR technical guidelines [8]. Even though IDSR system adoption by African countries was the most pragmatic approach given resource constraints, there is paucity of knowledge as to the vital recommendations emanating from assessing IDSR system functions [3]. Hence, this review focused on surveillance assessment studies

undertaken post-IHR (2005) adoption, which is in line with implementation of the revitalised IDSR system in Africa.

The health workforce across all health system levels are instrumental to achieving effective IDSR system implementation. Hence, giving due consideration to health personnel involvement and their perspectives on full optimisation of surveillance and response systems functionalities is vital to surveillance system improvement. There is insufficient review of literature on evaluation of key policy priorities based on health workers' perspectives ensuing from previous IDSR system assessment studies, which are pertinent to achieving communicable disease control in Africa. Previous systematic reviews have a limited focus on critically assessing fundamental recommendations derived from healthcare workers' perspectives on IDSR system improvement since adopting the revised IDSR guidelines in Africa [9,10]. Identifying recommendations derived from studies assessing the performance of IDSR system functions is key to focusing decision makers on the critical policy priority areas and guiding implementers towards improving disease surveillance and strengthening the overall health system. However, recommendations to strengthen specific surveillance functions needs to consider the unique nature of the diseases under surveillance. Therefore, the current review aimed to derive key recommendations resulting from IDSR system core, support and attribute functions assessment studies to improve implementation of the revitalised IDSR system in the African region based health workers' perspectives.

## Research question

What lessons can be learned from recommendations derived from previous IDSR system core, support and attribute functions assessment studies to improve implementation of the revitalised IDSR system in the African region based on health workers' perspectives?

The PICO (Population/Interest/Context/Outcome) framework [11,12] modified to suit qualitative review questions was used to identify keywords in the research question. Therefore, keywords used in the search strategy were derived based on the population comprising of healthcare workers and the phenomenon of interest was the integrated disease surveillance system encompassing core, support and attribute surveillance functions. Furthermore, the review context was Africa and the anticipated outcomes were recommendations to improving surveillance functions based on health workers' perspectives.

## Methods

A protocol for this systematic review was registered on July 1, 2019 in the International Prospective Register of Systematic Reviews (PROSPERO), registration number CRD42019124108 (S1 File). The review focused on literature published between 2010 and 2019 post-revised IDSR guidelines adoption in Africa. The search included published articles and grey literature for the period between 1st January 2010 to 31st January 2019. Systematic literature searches in PubMed, Web of Science and Cumulative Index for Nursing and Allied Health Literature (CINAHL) using keywords search strategy, in addition to manual reference searches were undertaken. Grey literature searched using World Health Organisation Library and Information Networks for Knowledge (WHOLIS). Each database was searched in consultation with the information specialist of the University of Pretoria Health Faculty Library. Keywords combination using Medical Subject Heading (MeSH) and free text terms relating to the IDSR system were used. The following keywords were used in various combinations ("surveillance", "public health surveillance" [MeSH], "integrated disease surveillance and response", AND "evaluation", "assessment" AND "health worker", "healthcare personnel" [MeSH], AND "Africa" [MeSH], "Sub Saharan Africa" (S2 File). Individual search terms were combined

using the appropriate Boolean operators to generate a list of citations that were saved into End-note X8 and screened for duplicates.

The review focused on deriving key recommendations based on IDSR system's core and support functions as has been defined by WHO as well as surveillance systems attributes as defined by Centers for Disease Control and Prevention (CDC) [13,14]. The inclusion criteria required the literature reviewed be; (1) published full text articles including unpublished studies and grey literature for the period between 2010 and 2019; (2) either quantitative or qualitative studies or both assessing implementation of one or more surveillance functions based on health workers' views through interviews and studies involving records reviews or observations; and (3) articles written in English language only. Exclusion criteria considered articles on surveillance assessment studies in countries outside Africa, articles published prior to 2010 before countries adopted the revised IDSR guidelines and articles written in any other language other than English (S1 Table).

## Data extraction and synthesis

All documents and published articles were manually reviewed with duplicates excluded. Preferred Reporting Items for Systematic Reviews and Meta-Analyses (PRISMA) Statement checklist for systematic reviews was utilised in the review process (S2 Table) [15]. Data extracted included the country of study, author's name, article publication year, country adoption year of IDSR revised guidelines, case disease/s of focus, study assessment methodology, surveillance functions assessed, key findings and recommendations (S3 File). Subsequently, the extracted data was entered into Microsoft Excel prior to analysis. Two reviewers (AKSN, RCK) undertook data extraction and discrepancies between the two resolved by consensus. Data synthesis for quantitative studies was conducted narratively [16]. Analysis of extracted data from qualitative studies was done using thematic synthesis [17]. A matrix of main themes of surveillance functions guided the thematic synthesis with emerging sub-themes (S3 Table). Main themes were based on pre-defined surveillance functions derived from the WHO assessment protocol for national disease surveillance systems and epidemic preparedness, the CDC updated guidelines for evaluating public health surveillance systems and the second edition guidelines for integrated disease surveillance and response in the African region [5,13,14]. On the other hand, emerging sub-themes were based on recommendations derived from the reviewed studies. Key recommendations from the reviewed studies were identified by first assessing the overall study conclusions then reviewing the study findings that informed the conclusions and recommendations. Reviewers preferred this approach since conclusions are derived from the main study findings, which are linked to critical recommendations that may bear policy implications.

## Literature quality appraisal

Dearholt and Dang's Johns Hopkins Nursing Evidence Appraisal Tool was used for quality appraisal of the reviewed literature. Quality of studies included was based on their strength of evidence (Level I-V) and quality of evidence (Grade A, B & C) (S4 File) [18]. This was done for each article included in the review by two authors (AKSN, RCK) answering a series of quality appraisal questions independently following which differences were discussed and a consensus reached on quality of literature to be included in the review. The strength of evidence was assigned level I, II, III, IV or V depending on whether the article was based on an experimental study, quasi-experimental study, non-experimental study, nationally recognised experts' opinion based on research evidence or individual expert opinion based on non-research evidence respectively. Furthermore, each included article was assigned grade A, B or C depending on

whether the quality of research evidence was of a high, good or low quality respectively [18]. Findings from articles considered to have lower levels of evidence or quality in contrast to findings of higher rated articles were not excluded from this review. However, results from these articles were assessed more critically.

### Risk of bias across studies

Majority of included studies except those supported by document reviews and observations depended on self-reporting by healthcare workers (HCWs). This may have biased their responses towards what they felt was socially desirable at the time of conducting the studies. Secondly, the review focused on assessment studies conducted in the African region, which may have limited the study's perspective from drawing lessons based on IDSR implementation outside Africa. The review only included studies written in English language, which may have introduced some form of selection bias. Lastly, the review was based on extracting relevant studies from four databases; hence limiting the search to what was available in these databases only.

## Results

### Summary of included studies

The systematic search cumulatively identified 7,491 records from all the databases including a manual reference search. Records retrieved included; 6,244 articles in PubMed, 1,084 articles in Web of Science, 124 articles in CINAHL, 26 grey literature records in WHOLIS and 13 manually searched references as described in the PRISMA flow diagram (Fig 1).

Abstracts of identified studies were reviewed and the full body text of selected articles read. All identified articles were written in English language. Of the 30 studies meeting the inclusion criteria, 28 (93%) were assessment studies involving health personnel interviews, 13 (43%) studies involved a combination of interviews and record reviews while 2 (7%) of the reviewed studies were exclusively based on records review. Surveillance assessment studies were based in 13 countries in the WHO-AFRO region (Ghana, Cameroon, Nigeria, Kenya, Ethiopia, Tanzania, Zimbabwe, Zambia, South Africa, Madagascar, Uganda, Sudan and Malawi). These countries adopted the revitalised IDSR guidelines between 2010 and 2016 [8]. The included assessment studies were based on the revised African IDSR technical guideline disease categories, with twenty studies focused on notifiable diseases requiring immediate reporting while three [19–21] out of the twenty studies mentioned diseases targeted for elimination and eradication including neglected tropical diseases (NTDs) such as guinea worm disease, trachoma and schistosomiasis. However, seven studies did not specify any particular disease in the assessment [22–28]. The reviewed studies covered a combination of surveillance functions with 24 focusing on core functions, 22 on support functions and 18 on surveillance attribute functions. Eighty-seven percent (26/30) of the reviewed studies adopted a cross-sectional study design with the remaining studies adopting either longitudinal [29], retrospective [30,31] or quasi-experimental [24] study designs. Furthermore, 63% (29/30) of studies in the review were solely based on quantitative data with two studies exclusively based on qualitative data. However, 30% (9/30) of the reviewed studies involved collection of both types of data. A summary of specific components covered under each of the surveillance function was undertaken, in addition to summarising findings from the reviewed assessment studies (Table 1). Moreover, recommendations to improve IDSR system implementation extracted from the included studies were summarised based on the surveillance functions (Table 2). Eighteen emerging sub-themes were derived from recommendations specific to four core functions and three support functions (Fig 2). Emerging sub-themes were the identified outcomes of strengthening specific

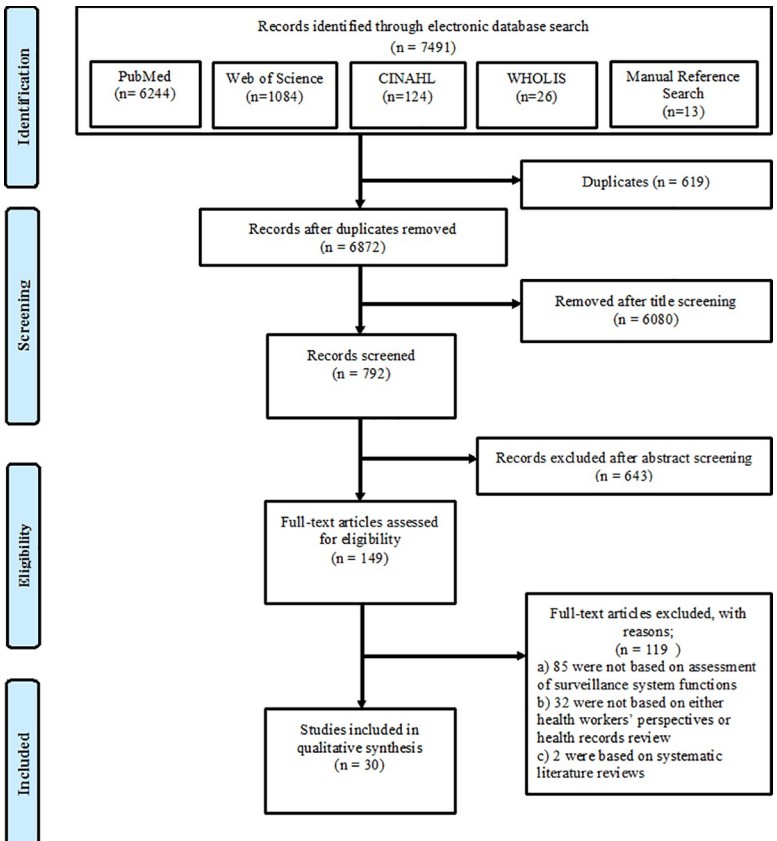

**Fig 1. Flow chart summarising the systematic review process.**

surveillance functions based on the recommendations. Sub-themes regarding resources were based on sub-categories of the different resource types. However, no specific sub-themes emerged from the surveillance attributes.

According to Dearholt and Dang's Johns Hopkins Nursing Evidence Appraisal Tool [18], most studies were assigned level III in terms of evidence strength since 28 out of 30 (93%) of the studies adopted non-experimental study designs (Table 1). In addition, three [28,32,42] studies were considered of low quality (Grade C) in terms of evidence quality considering their methodological approach. However, since these studies satisfied the inclusion criteria, they were included in the review and their study findings critically reviewed.

## Core functions

**Case confirmation.** Four of the 30 reviewed studies recommended strengthened case confirmation capacities [19,36,38,39]. Of these, two studies reported that prompt public health action requires enhanced laboratory capacity [19,36]. Laboratory services absence in health facilities to confirm cholera cases and outbreaks was reported in Cameroon [36]. Therefore, this required laboratory facilities and equipment be provided to ensure prompt detection, confirmation and response to cholera cases [36]. Similarly, laboratory capacity strengthening and prompt specimen collection was recommended in Ghana to ensure adequate surveillance and response preparedness to Ebola [39]. A sub-theme based on a recommendation derived from the reviewed studies alluded to improved specimen handling [38]. Functions relating to case

**Table 1. Literature summary and quality appraisal.**

| No. | Country | Year country adopted revised IDSR guideline | Case disease/s mentioned in the assessment study | Authors | Publication Year | Aim of the study | Assessment methodology | Surveillance system functions assessed | Key Findings | Evidence Levels[(a)] and Quality Ratings[(b)] |
|---|---|---|---|---|---|---|---|---|---|---|
| **1.** | Nigeria | 2013 | Not specified | Nnebue et al. [27] | 2013 | To determine reporting completeness and timeliness and ascertain the pattern of information transmission | Cross-sectional study; Multistage sampling; Sample size (270); Interviews, observation checklist and desk review | Core functions: Case registration, reporting, feedback Attributes: Data accuracy, reporting completeness and timeliness | Lack of IDSR reporting forms; Poor reporting completeness and timeliness | III, B |
| **2.** | Nigeria | 2013 | Diarrhoea, Measles | Abubakar et al. [32] | 2013 | To assess IDSR system implementation | Cross-sectional descriptive study; Multistage sampling; Interviews, records and reports review | Core functions: Reporting, feedback, data analysis Support functions: Standards and guidelines, resources | Poor IDSR implementation; Insufficient surveillance resources; Low feedback; Poor utilisation of standard case definitions | III, C |
| **3.** | Zimbabwe | 2012 | Cholera, Anthrax, Rabies | Maponga et al. [33] | 2014 | To evaluate the notifiable disease surveillance system | Descriptive cross-sectional study; Sample size (66); Interviews | Attributes: Acceptability, flexibility, simplicity, stability, data quality, timeliness, sensitivity, representativeness | Surveillance system was useful, acceptable, simple, sensitive and met reporting timeliness; Lack of reporting forms; Poor data quality; Limited feedback and low knowledge among health workers | III, B |
| **4.** | Nigeria | 2013 | Not specified | Nnebue et al. [26] | 2014 | To determine surveillance system functional status and examine the challenges faced across all surveillance levels | Descriptive cross-sectional study; Multi-sampling technique; Sample size (270); Interviews and observations | Core functions: Reporting, feedback Support functions: Supervision, training, resources | Lack of training; Poor health worker motivation; Inadequate supply of forms and other logistical support; Poor funding; Inadequate supervision and lack of prompt feedback | III, A |
| **5.** | Ghana | 2011 | Malaria, HIV/ AIDS, Cholera, Tuberculosis, Pneumonia, Meningitis, Poliomyelitis, Guinea Worm Disease | Adokiya et al. [21] (a) | 2015 | To evaluate IDSR system functioning and data quality | Observational study using mixed methods; Purposive sampling; Interviews and reports review | Core functions: Case detection, confirmation, reporting, data analysis, epidemic preparedness and feedback. Support functions: Supervision, training and resources | Low data quality; Poor case confirmation practices; Limited supervisory support; Ill-equipped laboratories; Poor feedback | III, A |

(*Continued*)

**Table 1.** (Continued)

| No. | Country | Year country adopted revised IDSR guideline | Case disease/s mentioned in the assessment study | Authors | Publication Year | Aim of the study | Assessment methodology | Surveillance system functions assessed | Key Findings | Evidence Levels[a] and Quality Ratings[b] |
|---|---|---|---|---|---|---|---|---|---|---|
| 6. | Ghana | 2011 | HIV/AIDS, Tuberculosis, | Adokiya et al. [34] (b) | 2015 | To assess the core and support functions of the IDSR system | Qualitative study; Convenience sampling; Sample size (18); Key informant interviews | Core functions: Case detection, registration, confirmation, data reporting, data analysis, epidemic response and feedback. Support functions: Supervision, training and resources | Increased reports submission; Enhanced data analysis; Improved human resource capacity; Inadequate supervision and training; Limited human and financial resources; Frequent staff turnover and poorly equipped laboratories | III, B |
| 7. | Zimbabwe | 2012 | Malaria, Rabies, Polio, Measles, Tuberculosis | Tsitsi et al. [35] | 2015 | To evaluate the notifiable disease surveillance system | Descriptive cross-sectional study; Purposive sampling; Sample size (53); Interviews | Attributes: Acceptability, usefulness, flexibility, simplicity, stability, sensitivity, data quality, representativeness and timeliness | The surveillance system was acceptable, flexible and simple but lacked stability, sensitivity and usefulness; Lack of reporting forms and guidelines; low knowledge among health workers | III, B |
| 8. | Ghana | 2011 | Ebola | Issah et al. [29] | 2015 | To assess the usefulness of the IDSR system in relation to Ebola | Longitudinal study design; In-depth interviews and documents review | Core functions: Case detection, case registration, case confirmation, reporting, epidemic preparedness and response. Support functions: Standards and guidelines, training, communication, coordination, resources, monitoring and evaluation. Attributes: Reporting timeliness | Low utilisation of Ebola standard case definitions; Poor registration; Adequate laboratory capacity; Inadequate training on Ebola epidemic preparedness | II, B |

(*Continued*)

**Table 1.** (*Continued*)

| No. | Country | Year country adopted revised IDSR guideline | Case disease/s mentioned in the assessment study | Authors | Publication Year | Aim of the study | Assessment methodology | Surveillance system functions assessed | Key Findings | Evidence Levels[a] and Quality Ratings[b] |
|---|---|---|---|---|---|---|---|---|---|---|
| 9. | Nigeria | 2013 | Not specified | Lar et al. [24] | 2015 | To assess challenges of IDSR system reporting | Quasi-experimental study; Random sampling; Sample size (108); Interviews and observations | Core functions: Reporting, feedback Support functions: Training | Increased health worker knowledge post-training; Increased training associated with reporting forms availability, recognition of reporting efforts and improved feedback | II, A |
| 10. | Cameroon | 2011 | Cholera | Ngwa et al. [36] | 2016 | To assess IDSR strategy activities focusing on Cholera | Cross-sectional study design; Sample size (30), Key informant interviews and documents review | Core functions: Case detection, reporting, outbreak detection and feedback Support functions: Standard guidelines, training, supervision, resources and laboratory capacity Attributes: Reporting completeness and timeliness | Lack of reporting equipment; Low data analysis and interpretation; Outdated cholera standard case definitions; Lack of well-equipped laboratories | III, B |
| 11. | South Africa | 2013 | 33 notifiable conditions in South Africa | Benson et al. [20] | 2016 | To determine key stakeholders perceptions on the notifiable disease surveillance system attributes | Cross-sectional survey; Interviews | Attributes: Acceptability, flexibility, simplicity, timeliness and usefulness | Low acceptability, flexibility and usefulness of surveillance system | III, B |
| 12. | Kenya | 2012 | 35 priority diseases as provided in the IDSR technical guideline | Mwatondo et al. [37] | 2016 | To determine the prevalence of adequate reporting and factors associated with IDSR reporting | Cross-sectional survey; Stratified random sampling; Sample size (183); Interviews and reports review | Core functions: Reporting Support functions: Standards and guidelines (i.e. case definitions), resources (i.e. computer hardware and internet) Attributes: Reporting timeliness and completeness | Sub-optimal reporting of priority diseases; Complete and timely reporting | III, A |
| 13. | Ghana | 2011 | Not specified | Adokiya et al. [22] | 2016 | To evaluate IDSR system reporting completeness and timeliness | Observational study design; Records review | Attributes: Reporting completeness and timeliness, data accuracy | Implementation of DHIMS2 showed improvements in IDSR weekly and monthly reporting data timeliness and completeness | III, B |

(*Continued*)

**Table 1.** (*Continued*)

| No. | Country | Year country adopted revised IDSR guideline | Case disease/s mentioned in the assessment study | Authors | Publication Year | Aim of the study | Assessment methodology | Surveillance system functions assessed | Key Findings | Evidence Levels[a] and Quality Ratings[b] |
|---|---|---|---|---|---|---|---|---|---|---|
| 14. | Sudan | 2013 | Meningitis | Baghdadi [38] | 2016 | To assess the core and support functions of the surveillance system with regards to meningitis | Cross-sectional study design; Interviews and observations | Core functions: Case registration and confirmation, reporting, feedback Support functions: Standards and guidelines (case definitions), training, laboratory capacity, communication facilities | Weak case confirmation; Inadequately trained health personnel; Poor feedback | III, B |
| 15. | Ghana | 2011 | Ebola | Adokiya and Awoonor-Williams [39] | 2016 | To assess the Ebola Virus Disease surveillance and response system | Observational cross-sectional study design; Sample size (47); Interviews | Core functions: Case detection and confirmation, reporting, feedback, outbreak preparedness Support functions: supervision, training, resources | Lack of case registers; Inadequate laboratory capacity | III, B |
| 16. | Zimbabwe | 2012 | Typhoid | Mairosi et al. [40] | 2016 | To evaluate the notifiable disease surveillance system | Descriptive cross sectional study design; Purposive sampling; Sample size (59); Interviews and records review | Core functions: Reporting Attributes: Usefulness, simplicity, acceptability, stability, flexibility sensitivity, data quality and timeliness | Low knowledge among health workers resulting to missed diseases, underreporting and poor case management; Surveillance system was unstable and lacked sensitivity | III, B |
| 17. | Nigeria | 2013 | Not specified | Iwu et al. [25] | 2016 | To identify gaps in disease reporting among health care workers | Descriptive cross-sectional design; Stratified simple random sampling; Sample size (449); Interviews and observations | Core functions: Reporting Support functions: Training, resources | Inadequate training; Lack of equipment and inadequate supply of reporting forms | III, A |
| 18. | Ethiopia | 2010 | Not specified | Begashaw and Tesfaye [28] | 2016 | To assess implementation of the IDSR system in health facilities | Descriptive cross sectional facility-based study; Multi stage sampling; Interviews | Core functions: Reporting, feedback, data analysis, Support functions: Resources | Limited data analysis; Non-functional equipment; and limited feedback from higher to lower levels | III, C |

(*Continued*)

**Table 1.** (*Continued*)

| No. | Country | Year country adopted revised IDSR guideline | Case disease/s mentioned in the assessment study | Authors | Publication Year | Aim of the study | Assessment methodology | Surveillance system functions assessed | Key Findings | Evidence Levels[a] and Quality Ratings[b] |
|---|---|---|---|---|---|---|---|---|---|---|
| 19. | South Africa | 2013 | Measles, Meningoccal Meningitis, Typhoid | Benson et al. [30] | 2017 | To compare laboratory surveillance with the notifiable diseases surveillance system | Retrospective study design; Records review | Attributes: Data quality, stability, representativeness, sensitivity and positive predictive value | Data incompleteness; Surveillance system lacked stability and representativeness | III, A |
| 20. | Zambia | 2012 | Dysentery, Malaria, HIV, Tuberculosis, Typhoid, Measles | Mandyata et al. [41] | 2017 | To investigate and report on the existing challenges in the implementation of the IDSR strategy | Qualitative study design; Purposive sampling; Key informant interviews | Core functions: Case detection, confirmation, registration, reporting, data analysis, response and control, feedback. Support functions: Training, logistical support, supervision. Attributes: Representativeness, stability | Availability of epidemic preparedness and response plans; Adequate human, technical and financial resources; Inadequately trained staff; Poor infrastructure and coordination challenges | III, B |
| 21. | Tanzania | 2011 | Malaria | Mboera et al. [42] | 2017 | To assess malaria surveillance system and explore the use of evidence in health planning and decision making at the facility and district levels | Cross-sectional study design; Purposive sampling; Sample size (20); In-depth interviews, observations and documents review | Core functions: Case registration, reporting, data analysis, response, feedback, Support functions: Standards and guidelines, training, resources, communication, Attributes: Reporting timeliness and completeness, usefulness | Poor data management; Inefficient reporting; Limited data analysis capacity; Over-burdened health staff; and weak communication systems | III, C |
| 22. | Ethiopia | 2010 | Vaccine Preventable Diseases i.e. Acute Flaccid Paralysis, Measles and Neonatal Tetanus | Lakew et al. [43] | 2017 | To assess the performance of disease surveillance and routine immunization | Cross-sectional study design; Purposive sampling; Interviews, observations and documents review | Core functions: Case confirmation, reporting, evaluation Support functions: Supervision, training, surveillance guidelines and case definitions, coordination | Lack of clear surveillance standard operating procedures; Limited active case searching; Incomplete case reports; Limited laboratory capacity | III, A |

(*Continued*)

**Table 1.** (*Continued*)

| No. | Country | Year country adopted revised IDSR guideline | Case disease/s mentioned in the assessment study | Authors | Publication Year | Aim of the study | Assessment methodology | Surveillance system functions assessed | Key Findings | Evidence Levels[a] and Quality Ratings[b] |
|---|---|---|---|---|---|---|---|---|---|---|
| 23. | Zambia | 2012 | Not specified | Haakonde et al. [23] | 2018 | To assess factors affecting IDSR system implementation in public health care facilities | Descriptive cross-sectional facility-based study design; Convenient sampling; Sample size (34); Interviews | Core functions: Reporting, feedback Support functions: Training, supervision, resources (logistical, financial, equipment) | Lack of periodical training and mentorship; Irregular supervision; Insufficient financial support; Lack of prompt feedback; and inadequate coordination and communication | III, B |
| 24. | Malawi | 2014 | Ebola, Tuberculosis, Malaria | Wu et al. [44] | 2018 | To describe the process of case identification and reporting in practice and explore the differences between the IDSR guideline and actual implementation | Mixed methods study design; Key informant interviews, focus groups and reports review | Core functions: Case detection, Reporting Support functions: Standard case definitions, Laboratory capacity, Training Supervision, Resources Attributes: Reporting completeness and timeliness | Differences between IDSR technical guideline and actual practice existed; System shortcomings resulted from financial constraints and poor infrastructure | III, A |
| 25. | Nigeria | 2013 | Cholera, Gastroenteritis, Measles, Typhoid fever, Schistosomiasis | Dairo et al. [19] | 2018 | To assess compliance with the surveillance and response guidelines for epidemic-prone diseases | Descriptive cross-sectional study design; Multi stage sampling; Sample size (198); Interviews, observations and records review | Core functions: Case detection, case confirmation, case registration, reporting, feedback, data analysis, epidemic preparedness Support functions: Standard case definitions, supervision, resources | Inadequate laboratory capacity at lower levels | III, B |
| 26. | Madagascar | 2013 | Malaria, Diarrhoea, Acute Respiratory Infections, Measles, Acute Flaccid Paralysis, Chikungunya | Randriami-arana et al. [45] | 2018 | To evaluate performance of the reinforced IDSR strategy using attributes and technological assessment | Evaluation study design; Random sampling; Interviews | Support functions: Standard and guidelines, resources (infrastructure) Attributes: Simplicity, data quality, completeness and timeliness | Improved IDSR data completeness; Poor timeliness and data quality | III, A |

(*Continued*)

**Table 1.** (Continued)

| No. | Country | Year country adopted revised IDSR guideline | Case disease/s mentioned in the assessment study | Authors | Publication Year | Aim of the study | Assessment methodology | Surveillance system functions assessed | Key Findings | Evidence Levels[a] and Quality Ratings[b] |
|---|---|---|---|---|---|---|---|---|---|---|
| 27. | Uganda | 2012 | Cholera, Polio | Masiira et al. [46] | 2019 | To present findings from an assessment of IDSR core activities and support functions | Cross sectional survey; Purposive sampling; Sample size (202); Interviews, focus groups and observations | Core functions: Case detection, case registration, case confirmation, reporting, feedback, data analysis, epidemic preparedness and response Support functions: Standard case definitions, training, resources, Attributes: Reporting completeness and timeliness | Inadequate training of health workers; Insufficient funding; Low perceptions on the IDSR system; Irregular supervision; High turnover of trained staff | III, A |
| 28. | Nigeria | 2013 | Measles | Ameh et al. [31] | 2016 | To evaluate the case-based measles surveillance system | Evaluation study; Retrospective records review; Interviews | Core functions: Case detection, case confirmation Attributes: Positive predictive value, data quality, acceptability, stability, representativeness, usefulness, timeliness | Sufficient case confirmation capacity; Declining reporting timeliness and positive predictive values; Surveillance system was useful and acceptable but lacked stability | III, B |
| 29. | Ghana | 2011 | Cholera | Adjei et al. [47] | 2017 | To evaluate the cholera surveillance system | Evaluation study; Records review; Interviews | Core functions: Case registration, data analysis, feedback Support functions: Resources Attributes: Simplicity, acceptability, stability, flexibility, usefulness, predictive value positive, sensitivity, timeliness, representativeness | Adequate case forms; Limited data analysis; Sufficient feedback; Adequate funding support; Surveillance system was simple, acceptable and flexible | III, B |
| 30. | Nigeria | 2013 | Cholera, shigella, measles, tuberculosis, hemorrhagic diseases, yellow fever, human influenza | Jinadu et al. [48] | 2018 | To determine the awareness and knowledge of health care workers about IDSR strategy for epidemic prone diseases | Cross-sectional facility-based study; Cluster sampling; Sample size (528); Interviews | Core functions: Case registration, reporting, Support functions: Training, resources Attributes: Simplicity | Reporting was simple but time consuming; Poor funding; Inadequate training and retraining of health workers; Limited human resource capacity and logistical support | III, A |

[a]**Evidence Levels:** Level I (Experimental studies, Randomised Controlled Trials); Level II (Quasi-experimental studies); Level III (Non-experimental studies).
[b]**Quality Grades:** A (High quality); B (Good Quality); C (Low Quality or major flaws).

**Table 2. Summary of key study recommendations from the reviewed studies.**

| Surveillance functions | Authors | Publication year | Country | Key study recommendations |
|---|---|---|---|---|
| **Recommendations on core functions** | | | | |
| **Case confirmation** | Ngwa et al. [36] | 2016 | Cameroon | Equipping health facilities and districts with surveillance personnel, computers and laboratories. Equipping health facility laboratories to ensure early detection, confirmation and rapid response. |
| | Baghdadi [38] | 2016 | Sudan | Formulating and distributing protocols on specimen (i.e. CSF) handling. |
| | Adokiya and Awoonor-Williams [39] | 2016 | Ghana | Improving laboratory capacity and prompt specimen taking. |
| | Dairo et al. [19] | 2018 | Nigeria | Strengthening laboratory support for disease surveillance at the health facility level. |
| **Reporting** | Nnebue et al. [27] | 2013 | Nigeria | Provision of reporting forms and other logistics on regular basis. Expanding sources of reporting. |
| | Lar et al. [24] | 2015 | Nigeria | Ensuring constant availability of IDSR reporting forms in the health facilities. |
| | Adokiya et al.(a) [21] | 2015 | Ghana | Addressing inconsistences between weekly and monthly surveillance data. |
| | Mwatondo et al. [37] | 2016 | Kenya | Providing urban settings with weekly reporting tools. |
| | Iwu et al. [25] | 2016 | Nigeria | Improved coordination, communication and support for disease reporting at local and state levels. |
| | Lakew et al. [43] | 2017 | Ethiopia | Improvement of surveillance documentation (i.e. copies of surveillance reports). |
| | Ameh et al. [31] | 2015 | Nigeria | Encourage all health facilities to be involved in reporting |
| | Jinadu et al. [48] | 2018 | Nigeria | Set up a good reward system to increase willingness for reporting |
| **Feedback** | Nnebue et al. [27] | 2013 | Nigeria | Ensuring adequate feedback of information. |
| | Abubakar et al. [32] | 2013 | Nigeria | Increased feedback from higher to lower levels. |
| | Nnebue et al. [26] | 2014 | Nigeria | Promptly providing feedback within the disease surveillance and notification system. |
| | Begashaw and Tesfaye [28] | 2016 | Ethiopia | Improved feedback from higher to lower levels. |
| | Benson et al. [20] | 2016 | South Africa | Provision of feedback to all key stakeholders in the Notifiable Disease Surveillance System. |
| | Mboera et al. [42] | 2017 | Tanzania | Providing feedback to motivate timely submission of reports. |
| **Data analysis** | Adokiya et al. [22] | 2016 | Ghana | Initiation of plans to scale up data entry in district health information management systems at the periphery level to ensure data accuracy. |
| | Mboera et al. [42] | 2017 | Tanzania | Capacity strengthening on data analysis and utilisation at the facility and district levels. |
| | Lakew et al. [43] | 2017 | Ethiopia | Data analysis and surveillance performance indicators monitoring at the zonal and district (woreda) levels. |
| **Recommendations on support functions** | | | | |

*(Continued)*

**Table 2.** (Continued)

| Surveillance functions | Authors | Publication year | Country | Key study recommendations |
|---|---|---|---|---|
| **Training** | Nnebue et al. [27] | 2013 | Nigeria | Periodic training and retraining of health personnel on disease surveillance and notification Regular in-house training of health workers. |
| | Nnebue et al. [26] | 2014 | Nigeria | Increased training for health workers on disease surveillance and notification. |
| | Issah et al. [29] | 2015 | Ghana | Improved training activities for health personnel. |
| | Tsitsi et al. [35] | 2015 | Zimbabwe | Improve healthcare workers' knowledge on notifiable disease surveillance systems through training. |
| | Lar et al. [24] | 2015 | Nigeria | Train health personnel on correctly filling and compiling IDSR reports. |
| | Ngwa et al. [36] | 2016 | Cameroon | Need for education and more supervision to ensure use of updated information and materials. Trained surveillance personnel at the district level will be a great boost to the IDSR strategy. |
| | Adokiya et al. [22] | 2016 | Ghana | Continued training of disease surveillance and health information officers to improve completeness, timeliness, data quality and accuracy of reporting. |
| | Benson et al. [20] | 2016 | South Africa | Additional training for all key stakeholders in the Notifiable Disease Surveillance System. |
| | Mwatondo et al. [37] | 2016 | Kenya | Training on IDSR system. |
| | Baghdadi [38] | 2016 | Sudan | Train laboratory and reporting unit health personnel. |
| | Mairosi et al.[40] | 2016 | Zimbabwe | Health workers to be trained on IDSR processes and the follow up actions. Simplify training material to ease understanding and improve knowledge levels. |
| | Iwu et al. [25] | 2016 | Nigeria | Regular health staff training programs. |
| | Mandyata et al. [41] | 2017 | Zambia | Addressing the challenge of inadequately trained human resources. |
| | Haakonde et al. [23] | 2018 | Zambia | Ensure resources are secured and made available towards the provision of regular IDSR trainings targeting health care workers engaged in IDSR implementation. |
| | Randriamiarana et al. [45] | 2018 | Madagascar | Healthcare staff require training on IDSR guidelines and case definitions. |
| | Masiira et al. [46] | 2019 | Uganda | Training of more health workers. Regular IDSR training. Incorporating IDSR training into the pre-service curriculum for health workers. Training of community members in IDSR. |
| | Adjei et al. [47] | 2017 | Ghana | Increased training and education on cholera transmission and prevention |
| | Jinadu et al. [48] | 2018 | Nigeria | Training to increase knowledge on use of IDSR forms |

*(Continued)*

**Table 2.** (Continued)

| Surveillance functions | Authors | Publication year | Country | Key study recommendations |
|---|---|---|---|---|
| **Supervision** | Nnebue et al. [27] | 2013 | Nigeria | Improved supervision for surveillance data collection and transmission. |
| | Nnebue et al. [26] | 2014 | Nigeria | Ensuring adequate supervision. |
| | Tsitsi et al. [35] | 2015 | Zimbabwe | Support and supervision to ensure notifiable diseases are notified using the correct channels. |
| | Ngwa et al. [36] | 2016 | Cameroon | Increased supervision to ensure use of updated information and materials. |
| | Lakew et al. [43] | 2017 | Ethiopia | Ensuring strict adherence to planned surveillance schedules (i.e. supervisory visits) |
| | Haakonde et al. [23] | 2018 | Zambia | Ensuring mentorship, regular and scheduled supervision is provided to strengthen IDSR implementation at the district level. |
| | Masiira et al. [46] | 2019 | Uganda | Strengthening supervision to improve IDSR performance. |
| **Resources** | Abubakar et al. [32] | 2013 | Nigeria | Provision of sufficient logistical resources and data management tools. |
| | Nnebue et al. [27] | 2013 | Nigeria | Improved funding for disease surveillance activities. |
| | Nnebue et al. [26] | 2014 | Nigeria | Provision of transportation, adequate supply of reporting forms and other logistics. |
| | Adokiya et al.(a) [21] | 2015 | Ghana | Ensuring adequate support for and communication within the IDSR system. |
| | Tsitsi et al. [35] | 2014 | Zimbabwe | Distribution of notification forms to all health facilities. Engaging telecommunication service providers to put up network boosters. |
| | Ngwa et al. [36] | 2016 | Cameroon | Equipping health facilities and districts with surveillance personnel, computers and laboratories. Computers and trained surveillance personnel at the district level will boost the IDSR strategy. |
| | Mwatondo et al. [37] | 2016 | Kenya | Designation of a dedicated surveillance focal person. Availing posters and guidelines on IDSR functions to improve weekly reporting. |
| | Iwu et al. [25] | 2016 | Nigeria | Adequate and equitable funding for the disease reporting process including regular staff remuneration. |
| | Begashaw and Tesfaye [28] | 2016 | Ethiopia | Ensuring sufficient surveillance resources are provided in health facilities. |
| | Mboera et al. [42] | 2017 | Tanzania | Strengthening the technical capacity of health facility, district and national levels on all aspects of health information systems. |
| | Mandyata et al. [41] | 2017 | Zambia | Addressing the challenge of poor infrastructure, coordination, lack of provision of optimal technical support to DHMTs and health facilities. |
| | Dairo et al. [19] | 2018 | Nigeria | Ensuring funds provision and other resources to surveillance workers to achieve effective disease control. |
| | Haakonde et al. [23] | 2018 | Zambia | Allocating funds to support IDSR activities in the health sector budget. |
| | Masiira et al. [46] | 2019 | Uganda | Increasing IDSR funding at district and health facility levels. |
| | Ameh et al. [31] | 2016 | Nigeria | Sustained provision for funding and increased logistical support |

(*Continued*)

**Table 2.** (Continued)

| Surveillance functions | Authors | Publication year | Country | Key study recommendations |
|---|---|---|---|---|
| **Timeliness and completeness** | Nnebue et al. [27] | 2013 | Nigeria | Increased awareness on importance of effective reporting. |
| | Maponga et al. [33] | 2014 | Zimbabwe | Availing information on when diseases are being notified. |
| | Issah et al. [29] | 2015 | Ghana | Improved timely reporting of notifiable conditions. |
| | Adokiya et al. [22] | 2016 | Ghana | Consistency during weekly and monthly reporting. Conducting further investigations to address reporting completeness. |
| | Mairosi et al. [40] | 2016 | Zimbabwe | Provision of time information on disease notification. |
| | Benson et al. [20] | 2016 | South Africa | Health reforms to address surveillance system reporting timeliness. |
| | Ngwa et al. [36] | 2016 | Cameroon | Enhancing human resource capacity. |
| | Mboera et al. [42] | 2017 | Tanzania | Awareness on specific reporting dates. |
| | Randriamiarana et al. [45] | 2018 | Madagascar | Reducing workload, increasing training and improving mobile network infrastructure. |
| | Masiira et al. [46] | 2019 | Uganda | Enhanced IDSR training and adopting mobile-based reporting. |
| **Data Quality/Accuracy** | Maponga et al. [33] | 2014 | Zimbabwe | Ensuring complete and precise reporting. |
| | Mairosi et al. [40] | 2016 | Zimbabwe | Reducing missed data occurrences. |
| | Adokiya et al. [22] | 2016 | Ghana | Adopting the DHIS2 reporting system starting from the peripheral level. |
| | Randriamiarana et al. [45] | 2018 | Madagascar | Providing quality control mechanisms to avoid transmission of erroneous data. |
| **Usefulness** | Benson et al. [20] | 2016 | South Africa | Health reforms to encourage use of surveillance data. |
| | Mairosi et al. [40] | 2016 | Zimbabwe | Documenting public health action reports based on surveillance data. |
| **Acceptability** | Maponga et al. [33] | 2014 | Zimbabwe | Provision of clear job descriptions. |
| | Tsitsi et al. [35] | 2015 | Zimbabwe | Aligning surveillance activities with day-to-day duties. |
| | Mairosi et al. [40] | 2016 | Zimbabwe | Designation of surveillance focal persons. |
| **Stability** | Maponga et al. [33] | 2014 | Zimbabwe | Availability of human resources. |
| | Tsitsi et al. [35] | 2015 | Zimbabwe | Availing material resources. |
| | Mairosi et al. [40] | 2016 | Zimbabwe | Enhanced training and provision of communication and logistical facilities. |
| | Mandyata et al. [41] | 2017 | Zambia | Improved internet connectivity and infrastructure. |
| | Benson et al. [30] | 2017 | South Africa | Provision of reliable diagnostic equipment. |

(Continued)

**Table 2.** (Continued)

| Surveillance functions | Authors | Publication year | Country | Key study recommendations |
|---|---|---|---|---|
| **Simplicity** | Maponga et al. [33] | 2014 | Zimbabwe | Ease of completing notification forms. |
| | Tsitsi et al. [35] | 2015 | Zimbabwe | Reduced length of time required to complete notification forms. |
| | Mairosi et al. [40] | 2016 | Zimbabwe | Ease of understanding surveillance system functionalities. |
| | Benson et al. [20] | 2016 | South Africa | Simplification of surveillance system at operational level. |
| | Randriamiarana et al. [45] | 2018 | Madagascar | Distribution and display of simplified and understandable terms of reference and case definition guidelines. |
| **Further recommendations** | | | | |
| **Alternative surveillance strategies** | Maponga et al. [33] | 2014 | Zimbabwe | Need for the Ministry of Health to develop an electronic based system for surveillance data reporting riding on the availability of mobile phone use. |
| | Tsitsi et al. [35] | 2015 | Zimbabwe | Adoption of an electronic/mobile channel in notifying diseases to cut down on costs of the paper-based system. |
| | Issah et al. [29] | 2015 | Ghana | Improving and focusing on community based surveillance system by bringing it into the mainstream surveillance for Ebola Viral Disease. |
| | Ngwa et al. [36] | 2016 | Cameroon | Equipping all health facilities with the 'green line' mobile surveillance approach. |
| | Benson et al. [20] | 2016 | South Africa | Introducing an electronic system including use of mobile telephone technology to address perceived weaknesses of the NDSS. |
| | Lakew et al. [43] | 2017 | Ethiopia | Properly formulating operational plans to improve active case search with realistic prioritization of visiting reporting sites. |
| | Wu et al. [44] | 2018 | Malawi | Improved technology infrastructure and adapting mobile technologies. Utilization of syndromic surveillance approach combined with systematic virological testing. |
| | Randriamiarana et al. [45] | 2018 | Madagascar | Improved data collection, compilation and transfer through an electronic system. Increasing mobile network coverage. |
| **Further research** | Maponga et al. [33] | 2014 | Zimbabwe | Need for further research on the effect of training health-care workers on the surveillance system. |
| | Adokiya et al. [22] | 2016 | Ghana | Further research to improve reporting completeness and timeliness of surveillance data. |
| | Mwatondo et al. [37] | 2016 | Kenya | Conducting further studies in rural or mixed settings in different Kenyan counties in order to gather information on the challenges of reporting in health facilities. |
| **Other recommendations** | Adokiya et al. 2015(b) [34] | 2015 | Ghana | Need to recognise disease surveillance activities as essential for the overall functioning of the health system. |

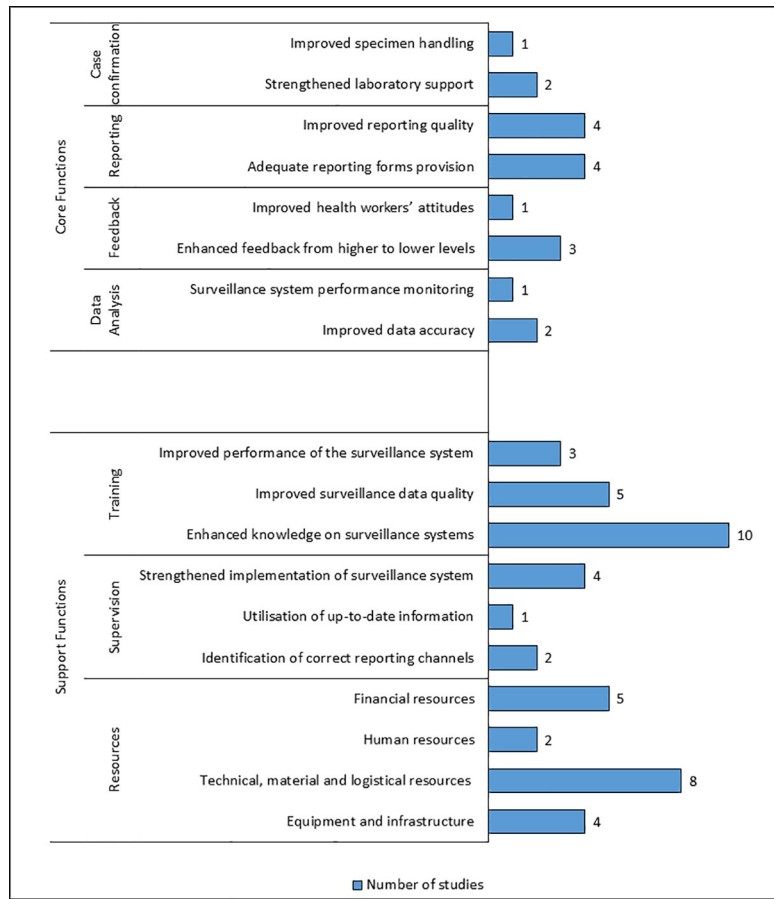

**Fig 2. Summary of themes, sub-themes and the number of reviewed studies.**

confirmation were absent in health facilities in Khartoum State, hence necessitating need to formulate and distribute protocols for specimen handling specific to meningitis [38].

**Reporting.** Slightly more than a quarter (8/30) of the reviewed studies provided recommendations on improving surveillance reporting [21,24,25,27,31,37,43,48]. Of these studies, two main sub-themes were identified on improved reporting quality [21,25,31,48] and adequate provision of reporting forms [24,27,37,43]. Health workers' awareness on the link between their day-to-day activities and disease reporting will improve their willingness to adhere to reporting guidelines [25]. A study in Kenya reported having weekly reporting forms present in health facilities significantly increased disease surveillance reporting odds [37]. Therefore, availing IDSR reporting tools would ensure continuity and consistency in reporting surveillance data [24,37]. In Ghana, inaccuracies and missing data in surveillance reports submitted from peripheral to regional level resulted from uncertainties on the most appropriate reporting channel [21]. This required addressing inconsistencies of weekly and monthly reports submitted through the various channels [21]. Advocating for improvements and clarity on the proper reporting channels would avoid frequent communication breakdowns and missing data in surveillance reports [21]. Improved surveillance documentation was recommended since most regional surveillance offices lacked active case searches written reports from reporting sites in Ethiopia [43]. An efficient reward system for reporting would motivate health personnel reporting efforts and involvement in the surveillance system [48].

**Feedback.** Up to 20% (6/30) of the studies recommended the need for improved feedback [20,26–28,32,42]. Further, two key sub-themes emanated from the reviewed studies on improving feedback, which identified the need for improved health workers' attitudes [20] and enhanced feedback from the higher to lower levels [28,32,42]. Feedback on reported data influences health worker's attitudes and willingness to participate in surveillance activities. However, inadequate feedback to health facilities may demotivate health workers, limiting their efforts towards efficient and timely reporting [42]. Health worker's low perceptions on the disease surveillance system's acceptability, flexibility and usefulness would be resolved through sufficient feedback [20]. Adequate feedback provision to motivate health workers to submit timely reports for malaria cases would address inefficient disease surveillance reporting in Tanzania [42]. Similarly, improved feedback from higher to lower levels would motivate health staff to report efficiently and influence their performance in surveillance activities as reported in Nigeria and Ethiopia [28,32]. Furthermore, ensuring adequate and prompt feedback within disease surveillance and notification system would alleviate major challenges faced within the system [26,27].

**Data analysis.** Of the studies (3/30) recommending for increased data analysis, one study indicated that data transmission challenges using paper-based reporting from periphery to district level increased error introduction likelihood in the reported data [22]. Hence, necessitating plans initiation for scaling up data entry in DHIMS2 at the periphery level to improve data accuracy in Ghana [22]. Similarly, challenges involving limited capacity and low evidence of proper data analysis at the hospital and district levels in Tanzania were to be mitigated by strengthening capacity for data analysis and availing tallying sheets, register books and reporting forms [42]. Further, limited use of outcomes from surveillance performance analysis in Ethiopia required an undertaking to analyse surveillance data and closely monitor surveillance performance indicators at regional levels [43]. The sub-themes derived from studies recommending routine data analysis were centered on surveillance system performance monitoring [43] and improved data accuracy [22,42].

## Support functions

**Training.** Sixty percent (18/30) of studies in the review recommended for enhanced training of health personnel. Three major sub-themes were derived from study recommendations regarding surveillance training and this included improved surveillance system performance [23,33,36], improved surveillance data quality [21,24,25,38,41] and enhanced knowledge on surveillance systems [26,27,29,35,37,40,45–48]. Low knowledge on correct forms for reporting notifiable diseases negatively affected timely disease reporting in Zimbabwe [35]. Therefore, health workers required training to improve their knowledge on notifiable disease surveillance systems through induction and on job training [35]. On-job training of health personnel during supervisory visits and sensitisation meetings is the common strategy applied or recommended especially at health facility level with limited formal training on IDSR implementation [29,33,35]. On the other hand, improved reporting practices as a result of forms availability and recognising health workers' reporting efforts was significantly associated with a post-basic training intervention in Nigeria [24]. Consequently, continued health worker training on correct form filling and reports compilation was recommended [24]. In Ghana, formal IDSR training with a focus on detection and reporting of Ebola Viral Disease (EVD) suspected cases was lacking at health facility and community levels [29]. Hence, an integrated and sustained funding support towards health personnel training at facility and community levels would ensure effective EVD suspected cases contact tracing and halt disease transmission [29]. Disease surveillance training especially at community, heath facility and district levels was limited

in comparison to training undertaken at the regional and national levels in Cameroon [36]. The mitigation measure recommended was to increase health personnel training at district level to enhance IDSR strategy implementation [36]. Previous studies conducted in West Africa recommended regular training of health staff to improve reporting and mitigate other challenges associated with inadequate training [25–27,47,48]. Furthermore, IDSR training was inadequate in Zambia resulting in health worker dependence on prior knowledge while executing their duties [41]. Therefore, they required adequate training to improve the quality and quantity of surveillance data being generated and utilised for decision-making [41]. Health workers' training needs on IDSR system aspects needed addressing to enable proper identification of designated focal surveillance persons in Kenya [37]. In South Sudan, increased health personnel training in hospital reporting units and laboratories would improve meningitis case-based reporting within the surveillance system [38]. Although heath workers in Zimbabwe perceived notifiable disease surveillance system to be simple and easy to use, training was necessary to increase their understanding of IDSR processes and follow up actions. This would be achieved through simplifying training materials to ease understanding of the system [40]. In Zambia, most respondents felt securing adequate funds to conduct periodical training and re-training could strengthen all IDSR system implementation aspects [23]. Similarly, challenges associated with inadequate training in Nigeria was to be mitigated through regular in-house training and re-training of health personnel on disease surveillance and notification [26,27]. Further, in Uganda, having an inadequate number of frontline health personnel trained on IDSR system was to be resolved through IDSR training incorporation in health worker's initial pre-service curriculum and community involvement in training [46].

**Supervision.** Seven out of the 30 studies indicated the need for supervision of surveillance activities [23,26,27,35,36,43,46]. The sub-themes relating to supervision that emanated from the study recommendations were based on strengthening implementation of the surveillance system [23,27,43,46], utilisation of up-to-date information [36] and identification of correct reporting channels [27,35]. Enhanced surveillance supervisory efforts at health facility level would ensure notifiable diseases are notified through correct channels [35]. However, most supervisory reviews only focused on surveillance activities involving immunisable diseases, tuberculosis and HIV/AIDs [35]. Similarly, partial supervision was undertaken in Cameroon at regional and district levels, while at community and health facility levels there was complete absence of supervisory activities [36]. Increased awareness on supervision benefits and efforts to enhance supervision would ensure utilisation of up-to-date surveillance information and materials amongst HCWs [36]. Surveillance focal persons irregularly provided supportive supervision for active case searches in Ethiopia, hence requiring strict adherence to planned surveillance schedules for conducting supervisory visits [43]. Furthermore, health workers at the district level in Zambia felt regular and scheduled supervisory assistance from higher levels would strengthen the IDSR system [23]. In addition, increased supervision was required to ensure disease notification systems were effective in data collection and information transmission in Nigeria [27]. Irregular supervision was still an existing challenge in the revitalised IDSR programme that required addressing to improve IDSR performance in Uganda [46].

**Resources.** Slightly more than half (16/30) of the studies identified the need for sufficient resources to support surveillance activities. Of the reviewed studies, recommendations on increased resource support for surveillance activities were focused on financial resources [23,25,27,31,46], human resources [36,37], materials and logistical support [19,26,28,31,32,35,37,42] and equipment and infrastructure [21,35,36,41]. Surveillance data analysis and management tools unavailability at health facility and district levels was reported in Cameroon [36]. Hence, requiring health facilities and district levels to be equipped with computers [36]. Similarly, data management tools availability was to be complemented by

their functionality to ease surveillance data entry and analysis [28,32]. On the other hand, the main challenge facing cellphone communication reporting channels was poor network infra- structure in Zimbabwe [35]. Hence, requiring telecommunication service providers engage- ment to set up network boosters to improve communication and timely reporting [35]. Limited utilisation of routine health information for performance monitoring was to be miti- gated through health information systems strengthening at all surveillance levels in Tanzania [42]. Notification forms unavailability in Zimbabwe hindered HCWs efforts for disease notifi- cation and delayed epidemic investigations [35]. Therefore, this necessitated the distribution of notification forms to all health facilities [35]. In Kenya, health facilities displaying visual aids for IDSR functions were more likely to report surveillance data [37]. Hence, to strengthen these efforts it was recommended that posters and guidelines on IDSR functions be provided to improve reporting [37]. In Zambia, IDSR technical guidelines were unavailable in health facilities, hence they lacked the appropriate procedures for handling suspected cases of notifi- able diseases [41]. This identified the need for technical support especially at health facility lev- els to promote and improve early disease detection [41].

Health facilities lacking health workers designated to manage disease surveillance data had decreased odds of adequate reporting [37]. Hence, designating a surveillance focal person in health facilities would improve surveillance reporting [37]. Likewise, there was need to equip the district and health facility levels with trained surveillance personnel in Cameroon [36]. Healthcare personnel identified lack of financial aid as a hindrance to IDSR implementation in Zambia. Hence, necessitating funds allocation in the health sector budget to support IDSR activities [23]. In Nigeria, improved funding would ensure effective surveillance data collection and transmission process [27,31]. Furthermore, in the South Eastern State of Nigeria, adequate and equitable funding was required to facilitate the disease reporting process [25]. Similarly, increased IDSR funding was recommended to support surveillance activities at the district and health facility levels in Uganda [46].

## Surveillance attributes

Key recommendations on improving the surveillance attributes were specific to reporting timeliness and completeness, data quality and accuracy, usefulness, acceptability, simplicity and stability of the surveillance system [20,22,30,33,35,36,40–42,44–46].

**Timeliness and completeness.**   Thirty-three percent (10/30) of the reviewed studies rec- ommended improved reporting timeliness and completeness. An under-performing surveil- lance quality function requiring improvement was lack of timely reporting within the 24 hour window period for Ebola suspected cases at the regional surveillance unit in Ghana [29]. Simi- larly, inconsistencies in weekly and monthly reporting timeliness were observed in northern Ghana, hence requiring an urgent need to strengthen the disease surveillance system to enable rapid response to infectious disease outbreaks [22]. Information unavailability on disease noti- fication from health facilities to district level, negated efforts to ascertain surveillance data timeliness in Zimbabwe [33,40]. Similarly, HCWs in Tanzania responsible for malaria surveil- lance data reporting were unaware of specific dates when reports were submitted from health facilities to the next reporting level [42]. Therefore, information provision bearing specific reporting dates would be critical to determining surveillance system reporting timeliness. Improved reporting timeliness and completeness in Uganda resulted from enhanced IDSR training, which created increased disease surveillance awareness amongst health providers coupled with mobile-SMS based reporting [46]. Similarly in Malawi, adapting an electronic reporting system and mobile technologies would mitigate disease notification challenges from health facilities to the next level [44]. Furthermore, increased awareness on effective reporting

would resolve reporting reluctance amongst health workers and improve reporting timeliness in Nigeria [27]. Reduced workload, enhanced training and improved mobile infrastructure would improve reporting completeness and timeliness according to medical and paramedical agents in Madagascar [45]. Moreover, few healthcare providers in South Africa confirmed that the existing notifiable disease surveillance system (NDSS) provided timely information to prompt action. Hence, there were calls for future reforms to give priority to 'timeliness' attribute of South African NDSS to ensure effective disease outbreak containment [20]. Variations in reporting completeness across health system levels in northern Ghana and Cameroon were to be mitigated through undertaking further research investigations to address the root causes and enhancing human resource capacity respectively [22,36].

**Data quality and accuracy.** Four of the 30 studies indicated the need to ensure surveillance data accuracy [22,33,40,45]. Scaling-up data entry into the District Health Information Management System (DHIMS2) starting from the health system periphery level would address data quality and accuracy concerns [22]. Erroneous data transmission across surveillance system undermined data quality and surveillance data reliability [45]. Likewise, ensuring missing data in disease notification forms occurred less frequently would improve reported data quality [33,40]. Moreover, data entries completeness and correctness in notification forms was a data quality measure in Zimbabwe [40].

**Simplicity.** Up to 17% (5/30) of the studies required simplification of the surveillance systems [20,33,35,40,45]. Availing easily understandable and simplified terms of reference and case definitions would ease surveillance activities in Madagascar [45]. Notifiable disease surveillance system evaluation in Zimbabwe identified ease and duration of completing disease notification forms as a determinant of system's simplicity [33,35,40]. Health workers' positive perceptions on simplicity of the system motivates their involvement in notifying diseases [40]. In contrast, healthcare stakeholders involved at operational level of the surveillance system in South Africa perceived the system to be complex compared to their counterparts at health management level. Hence, identifying need for simplifying the system at disease detection and response level [20].

**Usefulness.** Two studies in the review alluded to usefulness of existing surveillance systems [20,40]. There were calls for reforms of South African NDSS to encourage surveillance data use by healthcare providers for outbreak response and communicable diseases control [20]. Besides, effective reports documentation on public health actions or decisions following data collected through disease surveillance systems would be vital in assessing system's usefulness [40].

**Acceptability.** Three of the 30 studies gave recommendations on acceptability of the surveillance system [33,35,40]. To resolve health workers' lack of willingness to notify diseases in Zimbabwe, there was need for clear designation of surveillance focal persons within health facilities [40]. Similarly, health workers' willingness to participate in surveillance activities was influenced by disease surveillance being in line with their job description [33,35].

**Stability.** Sixteen percent (5/30) of the studies identified the need for stable surveillance systems [30,33,35,40,41]. Enhanced stability of the existing surveillance system in Zimbabwe required an increased number of staff are trained on disease surveillance and provision of functional communication equipment and transport facilities [40]. Similarly, reports on surveillance systems evaluation in Zimbabwe assessed system's stability based on human and material resource availability [33,35]. Improved stability of NDSS in South Africa implied the system should be able to provide reliable diagnostic results on notifiable diseases [30]. Furthermore, surveillance system stability in Zambia was dependent on internet connectivity consistency or internet outages frequency for a specific time during reporting [41].

## Alternative surveillance strategies

In studies conducted in Zimbabwe, the reporting process was cost intensive due to transport expenses incurred while submitting paper-based disease notification forms. Therefore, electronic-based system adoption for reporting within health facilities would minimise these costs [35,40]. Disease notifications using the paper-based system for sending notification forms was cost intensive. Hence, requiring the establishment of an electronic-based system for surveillance data reporting and mobile phone technology utilisation [33]. Similarly, there was need to equip health facilities with mobile phone surveillance to effectively capture cholera cases in Cameroon [36]. A suspected Ebola outbreak in Ghana necessitated community-based disease surveillance revival as an active mainstream surveillance system to effectively detect and monitor suspected diseases [29]. In Ethiopia, operational plans formulation for conducting prioritised surveillance visits to specific reporting sites would improve active case searches [43]. To resolve discerned weaknesses in attributes of NDSS in South Africa, it was important to establish an electronic surveillance system utilising mobile phone technology [20]. Similarly, adopting mobile technologies in addition to utilising syndromic surveillance approaches were recommended to strengthen IDSR system in Malawi [44].

## Further research on surveillance

Inconsistencies in surveillance data completeness and timeliness in northern Ghana necessitated further research to mitigate this shortcoming [22]. Limited knowledge amongst health workers on the NDSS coupled with its sub-optimal performance was reported in Zimbabwe [33]. The resolution reached was to initiate further research efforts to assess the effect of health worker training on surveillance system performance [33]. In Kenya, further studies to establish reporting challenges facing health facilities in remote settings were recommended [37]. Similarly, there was need to undertake further research in Zimbabwe to ascertain the effect of training health workers on surveillance system aspects [33].

Noteworthy, sub-themes frequency effect size summary based on how often a particular sub-theme appeared in the body of literature reviewed indicated sub-themes relating to knowledge on surveillance systems; technical, material and logistical resources; financial resources and improved surveillance data quality as the predominant emerging sub-themes with frequency effect sizes of 33.3%, 20%, 20% and 16.7% respectively. On the other hand, intensity effect size of studies based on how much each study contributes, in terms of the number of sub-themes it included to the overall body of literature reviewed indicated articles authored by Ngwa et al. 2016 (27.8%), Nnebue et al. 2013 (27.8%), Tsitsi et al. 2015 (22.2%), Lakew et al. 2017 (22.2%) and Baghdadi, 2016 (22.2%) contributed significantly to the reviewed literature [27,35,36,38,43] (S4 Table).

## Discussion

The reviewed disease surveillance assessment studies clearly indicate milestones achieved since adopting the revised IDSR guidelines in Africa, in comparison to findings from a previous review [9]. However, the current review still identifies persistent challenges in IDSR system implementation. This review of recommendations ensuing from prior studies focused on assessing IDSR system functions based on a matrix of major themes inclined to specific surveillance core, support and attribute functions. From the review, it was apparent that most disease surveillance assessment studies conducted in Africa mainly advocated for health worker training [20,22–27,29,33,35–38,41,46]. Training supports and cuts across strengthening all surveillance functions. Moreover, knowledge impartment through training on IDSR system functions was considered a critical strategy to ensure disease surveillance system effective

functioning. The review identified feedback and reporting as essential surveillance core functions while training, resources and supervision as vital surveillance support functions as perceived by HCWs in Africa. The predominant recommendations regarding surveillance attributes focused on timeliness and completeness. The recommendations aimed to influence existing health policies by strengthening IDSR system critical functions parallel to reinforcing core surveillance capacities laid out in the IHR [3].

Case confirmation as core function is paramount to informing effective and prompt action to alleviate disease outbreaks. Therefore, provision of fully functional and adequately equipped laboratory facilities right from the peripheral level is critical for surveillance system improved performance. In line with our findings, IDSR implementation consolidates surveillance efforts with laboratory support to achieve effective public health action and response [29]. However, laboratory capacity challenges relating to limited supplies and low knowledge on specimen handling still exist in Africa despite the progress made in complying with IDSR and IHR requirements [9,49,50]. A key policy challenge relating to laboratory capacity among African countries is lack of ownership and consideration of laboratory undertakings and budgets in national health plans [49]. Hence, limiting resource mobilisation and sustainability of laboratory capacity.

Most health systems in Africa especially at peripheral levels rely entirely on paper-based reporting mechanisms due to limited technological and infrastructural capacity [22]. IDSR implementation in Africa is still being confronted with reporting challenges especially at health facility level, which is characterised by limited generation of reliable health information [25]. In addition, effective disease surveillance is difficult to achieve in a health system with inadequate infrastructure and a limited health workforce encumbered with surveillance data reporting processes [21]. Similar to our study findings, reporting forms and guidelines unavailability has an impact on health workers reporting performance and impedes their ability to conduct outbreak investigations [33]. In addition, health workers' awareness on the link between their day-to-day activities and disease reporting improves their willingness to adhere to reporting guidelines [25,35,37]. Furthermore, reporting forms missing critical information might upset data analysis efforts and further investigations [33]. Hence, the overall surveillance data quality as depicted by current study findings dictates public health response quality.

Feedback is an essential surveillance function and a core IDSR indicator in measuring system's performance [51]. Reviewed studies identified verbal feedback to health facilities as common practice by health personnel usually through half year or quarterly meetings [21,33,36]. Further, the review identified a major challenge in feedback mechanisms of existing surveillance systems in Africa, which neglect peripheral levels [36]. Comparably, limited feedback especially at the lower levels have previously been reported, which may demotivate health worker involvement and attitudes towards disease surveillance activities [50,52]. The current review identified recommendations to mitigate inconsistent feedback to lower levels resulting from absence of formal feedback plans and mechanisms as reported in other studies [52].

Analysed data enables monitoring of disease trends to inform case-based investigations and response [36]. However, minimal and basic data analysis is a common practice in health facilities with little or no documented evidence of analysed data as was evident from the reviewed study findings [42]. This result from misguided perceptions on the purpose of generating surveillance data for onward reporting to higher levels rather than utilisation at source [19]. Minimal routine data analysis especially at the lower level facilities has been attributed to limited knowledge and skills among health workers to undertake analysis of surveillance data and absence of simplified guidelines as suggested by some authors [51–53].

In line with findings from the review, regular health personnel training is linked to strengthened surveillance systems through improved reporting quality and enhanced

supervision and feedback across surveillance levels [9]. Low knowledge on the surveillance system among health personnel due to infrequent training is considered a key factor affecting IDSR implementation and overall performance [23,51]. Similar to the current review findings, training enhances health worker knowledge on surveillance system, data accuracy and improve reporting timeliness and completeness [21,22,54]. However, frequent turnover of trained health staff has a limiting effect on IDSR system optimal functioning [21].

Periodic supervision influences reporting frequency and the quality of surveillance data being reported [21]. From the reviewed studies, it was evident that correct identification of reporting channels was dependent on regular supervision. Therefore, recognising the critical role played by supervision of surveillance activities [36]. Supervisory activities lack consistency with efforts mostly initiated during disease outbreaks and this poses a major challenge to achieving effective IDSR implementation and performance [41,46]. Furthermore, strengthened and well-performing surveillance systems could be achieved through increased supportive supervision by adapting formalised supervisory plans [9,43].

Adequate resource provision facilitates IDSR system optimal functioning. The IDSR strategy was founded on the principle of utilising scarce healthcare resources to effectively achieve disease prevention and control. This review identified resource challenges relating to unavailability of reporting tools, lack of technical guidelines and over dependence on paper-based reporting mechanisms [22,36,41]. Likewise, a preceding review linked inadequate electronic equipment and unavailability of information, education and communication materials and job aids to IDSR system sub-optimal performance [9]. Moreover, the main factors contributing to low quality surveillance data generation are attributed to inadequate funding, limited human resource capacity and unavailability of supporting materials [34,55].

In the pre-adoption phase of revised IDSR guidelines, sensitisation and health personnel training would aid improved reporting timeliness and completeness [56]. Similarly, post-adoption of the revised IDSR guidelines identified enhanced health worker training as a strategy for improved reporting [45,46]. However, infrastructural constraints relating to logistical and communication systems negatively impact reporting timeliness [57]. Hence, calls for designing and adapting electronic or mobile reporting systems are justified [19,20,33,44,45,58,59].

Evidently, of the reviewed studies, only a few assessed the existing surveillance system considering NTDs. For instance, health workers in Madagascar were more aware of case definitions for common conditions such as malaria, diarrhea and respiratory infections compared to other neglected conditions like dengue fever [45]. This low awareness resulted from lack of case definition guidelines, terms of reference and inadequate IDSR training, hence influencing surveillance system's simplicity and applicability to other neglected conditions [21,45,50].

The review further identified pertinent recommendations to achieving improved surveillance performance through influencing health personnel perceptions towards surveillance attributes. Effective disease surveillance systems performance depends on ease of understanding system's functionalities [33,35,40,60]. Elsewhere, perceived surveillance data usefulness was lower amongst healthcare stakeholders responsible for disease detection and response in comparison to those in health management overseeing surveillance activities [20]. An acceptable surveillance and response system is well defined by health workers' willingness to voluntarily participate in surveillance activities [14]. Additionally, the functioning state of surveillance core and support functions for instance case confirmation, training, human resources, equipment and communication infrastructure may influence surveillance system's stability [30,33,35,40,41].

Other recommendations in the studies reviewed focused on alternative surveillance strategies. Efforts for active case searching at peripheral levels can be enhanced through establishing well-structured community based disease surveillance systems [29]. Furthermore, effective

active case searching could be achieved through properly designed operational plans targeting priority surveillance areas with high disease reporting sites [43]. Further assessment studies are required in remote settings to strengthen the IDSR system at the peripheral levels bearing higher disease burdens [37]. Research efforts initiation to address key challenges affecting IDSR system implementation will ensure surveillance system core, support and attribute functions optimal performance in Africa.

Our study had a couple of limitations. First, the review included articles written in English language only, which may have led to some degree of selection bias. Secondly, the reviewed studies were extracted from only four databases and the review might have missed other studies; however, we believe the search was able to comprehensively capture the surveillance assessment studies conducted in the African region within the selected period. Thirdly, findings were drawn from responses that may have been influenced by social desirability among study participants. Therefore, surveillance assessment studies conducted in future could incorporate observations and document reviews to limit self-reporting bias. Lastly, future reviews could assess and draw lessons on improving IDSR implementation from studies conducted outside the African continent.

## Conclusion

Evidently from this review, consolidated efforts to strengthen all strategic IDSR components is cardinal to achieving effective IDSR strategy implementation in Africa [51]. Notably, the reviewed studies prioritised surveillance systems assessment with regard to notifiable diseases. However, there was limited focus on other diseases of public health importance such as neglected tropical conditions. The review illustrated that implementation of key recommendations based on health workers' perspectives will prioritise use of scarce healthcare resources to strengthen specific surveillance system functions. Furthermore, health policy reviews with a keen focus on strengthening surveillance reporting, feedback, supervision, health worker training, resources and reporting timeliness and completeness could achieve effective IDSR system implementation especially at lower surveillance levels. In the future, it would be pertinent for the WHO Regional Office for Africa in collaboration with national health ministries to undertake periodic surveillance assessment studies tailored to local settings for improved IDSR system implementation and performance.

## Supporting information

**S1 Table. Excluded studies from systematic literature review.**
(DOCX)

**S2 Table. PRISMA checklist.**
(DOCX)

**S3 Table. Surveillance system functions.**
(DOCX)

**S4 Table. Summary of sub-themes with frequency and intensity effect sizes.**
(DOCX)

**S1 File. PROSPERO protocol.**
(PDF)

**S2 File. PubMed search strategy.**
(DOCX)

**S3 File. Data extraction form.**
(DOCX)

**S4 File. Johns Hopkins nursing evidence-based practice appraisal tool.**
(DOCX)

## Acknowledgments

We acknowledge Mrs. Estelle Grobler (Information Specialist) in the Faculty of Health Sciences Library, University of Pretoria for her assistance in the literature search exercise.

## Author Contributions

**Conceptualization:** Arthur K. S. Ng'etich.

**Data curation:** Arthur K. S. Ng'etich, Ruth C. Kirinyet.

**Formal analysis:** Arthur K. S. Ng'etich, Ruth C. Kirinyet.

**Investigation:** Arthur K. S. Ng'etich.

**Methodology:** Arthur K. S. Ng'etich.

**Supervision:** Kuku Voyi, Clifford M. Mutero.

**Validation:** Arthur K. S. Ng'etich.

**Writing – original draft:** Arthur K. S. Ng'etich, Clifford M. Mutero.

**Writing – review & editing:** Arthur K. S. Ng'etich, Kuku Voyi, Ruth C. Kirinyet, Clifford M. Mutero.

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
