## [Decision Letter · Decision Letter 0]

26 Jun 2020

PONE-D-19-27712

A systematic review on improving implementation of the revitalised integrated disease surveillance and response system in the African region: a health workers' perspective

PLOS ONE

Dear Dr. Ngetich,

Thank you for submitting your manuscript to PLOS ONE. After careful consideration, we feel that it has merit but does not fully meet PLOS ONE’s publication criteria as it currently stands. Therefore, we invite you to submit a revised version of the manuscript that addresses the points raised during the review process.

We look forward to receiving your revised manuscript.

Kind regards,

Jianhong Zhou

Associate Editor

PLOS ONE

Journal Requirements:

Reviewers' comments:

Reviewer's Responses to Questions

**Comments to the Author**

1. Is the manuscript technically sound, and do the data support the conclusions?

Reviewer #1: Partly

Reviewer #2: Yes

2. Has the statistical analysis been performed appropriately and rigorously? 

Reviewer #1: N/A

Reviewer #2: N/A

3. Have the authors made all data underlying the findings in their manuscript fully available?

Reviewer #1: Yes

Reviewer #2: Yes

4. Is the manuscript presented in an intelligible fashion and written in standard English?

Reviewer #1: Yes

Reviewer #2: Yes

5. Review Comments to the Author

Reviewer #1: Thank you to the authors for producing this detailed and interesting review of recommendations for improving the implementation of disease surveillance in Africa, with a particular focus on system and organisational issues from the perspective of health workers. I am sure this review will be of interest and useful to those working in disease surveillance internationally, identifying common challenges in implementing these systems and recommendations for their improvement, while providing a collated set of evaluation case studies.

I found the review relatively straightforward and clear, although I do have some suggestions to improve the clarity of the methods. Please note that my expertise is in systematic review methodology, and not in disease surveillance systems, so some of my comments may arise from my lack of knowledge of common concepts and terminology in this field.

Introduction, last para (p4, line 93 onward):

• It’s not clear to me what the difference is between the focus of the previous reviews, and the focus of this one. Perhaps this could be more specifically described? If the difference is the healthcare work perspective, perhaps this could be stated more simply. To me, the following descriptions sound the same:

o “critically assessing fundamental recommendations arising from assessment studies regarding IDSR system improvement”

o “key recommendations resulting from IDSR system core, support and attribute functions assessment studies”

Table 1 (p5, line 110):

• It might be helpful to write this table using plain language sentences, rather than search terms, to help people understand quickly and easily what you were including. Especially as these terms aren’t necessarily those used in your final search, as included in your supplementary files.

Eligibility criteria (p7, line 135)

• These criteria could be written more clearly. Make sure it’s clear whether you’re including:

o Only assessments of performance of the IDSR or assessments of any aspect of disease surveillance

o Health workers’ views on the implementation & operation of the system (i.e. studies that explicitly ask health workers to express their views e.g. through surveys or interviews), descriptive characteristics of the implementation and operation of the system that are relevant to health workers (e.g. that could be observed by researchers or policy staff rather than health workers), or any other outcomes (e.g. stats on the effectiveness of surveillance systems or disease outcomes)

• I’m not sure what the difference is between the studies that meet the main eligibility criteria, and the additional studies that " studies involving record reviews or observations to assess disease surveillance and response systems”. How are these studies different from your included studies, and how are their results treated differently in the review? If they’re not treated differently, then they can be included in the main eligibility criteria. I think you explain this more clearly in your description of include studies in the Results section.

Search

• The search terms included in the text are not the same as those used in the actual search strategy included in the supplementary documents. Could this be corrected?

• If I were drafting this search, I might expect to use some terms such as specific country names (as not all abstracts will specifically refer to Africa), as well as terms relating to evaluation or assessment. I might also consider searching the African Index Medicus. However, I know you have developed this search in collaboration with an information specialist, so you might have made an expert judgement that these would not improve the search.

Data extraction (p.7 Line 146)

• The PRISMA statement checklist is a very brief checklist of the presence or absence of methods in a systematic review, and cannot be used for data collection from primary studies. This text should be removed, and a description added of what kinds of data were collected. For example, in your PROSPERO record, you note that “The data to be extracted will be as follows: author's name, article publication year, adoption year of IDSR revised guidelines, target disease/s, study assessment methodology, surveillance functions assessed, key findings and recommendations.” Ideally, a copy of your data collection form could be included as a supplementary document, if available.

• Later in the review, in the discussion (p.52, line 478), you discuss developing a matrix of subthemes based on a previous review. I would love to see this discussed here, in the methods section, to describe how you identified themes in the included studies and categorised them according to themes. You could include your matrix as a table or figure. Also, did you add any new items to the matrix that you found in your included studies?

• It would be helpful to be more specific about exactly what kinds of outcomes you were looking for – that is, outcomes relating to the implementation and operation of the IDSR, but not measures of actual disease surveillance or disease outcomes in the population?

• It would be very helpful to provide some additional detail on how you’re identifying which are the key problems identified in each study, and which are the key recommendations. Did you collect every identified problem and every recommendation identified that matched your matrix? Did you select those that you thought were the most important? Did you only select those that matched your original matrix? This is very important so your readers can see how you made your choices and helps them trust your findings.

• The reason I’m asking for more detail is not because I think the review was not done appropriately, but just to increase transparency and replicability – e.g. if I were a new research assistant, could I follow the instructions in this methods section and successfully replicate the review?

Quality appraisal

• The Nursing Research Appraisal tool used to appraise the included studies is not a true risk of bias assessment tool, and it includes elements that do not relate to study bias, e.g. clarity of writing. However, in the context of this review, which aims to map diverse recommendations, rather than identify a correct effect estimates, I think a more general tool is probably fine.

Results

• Could you include a summary of the study designs found (e.g. cross-sectional, qualitative)

• Were there any subthemes identified in your matrix for which no data were found? i..e that you thought might be issues of concern, but weren’t?

• Excluded studies – it would add to the transparency of the review if you could provide a list of excluded studies, with their reasons for exclusion, in addition to the PRISMA flow diagram. This could be, for example, either a full list of studies screened in full text, or a subset of these that readers might be most likely to think

Table 4 (p48)

• This table would be easier to read if it were a frequency histogram or bar chart, showing a bar indicating the number of studies with that subtheme, rather than listing all the studies individually. It might even fit on a single page.

• Should this table be linked from the last para on p.46 (line 455), instead of Table 3? I would recommend it is linked from the Results section, and not first introduced in the Discussion section.

Discussion

• Should this reference to Table 4 actually be a reference to Table 3? (p47, line 465)

• The summary of the results in the Discussion could be more brief – it’s currently almost as long as the Results section (9 pages compared to 10). If there are details here that are important to interpreting the results, could they be included in the Results section? This would then allow the Discussion to be more of a brief overview of main findings, rather than repeating so much detail that might overwhelm readers. This would also make space to include some more discussion of your findings overall.

• The Discussion could also include some mention of the strengths and limitations of the review design (e.g. those mentioned in the Risk of bias across studies section).

• Will some recommendations be easier/harder to implement (e.g. some requiring significant additional resources which may not be available, others involving organisational change that’s more achievable?)

• Who is this data for? Who can or should act on it?

Conclusion

• You noted in the Conclusions that you felt there was not much evidence in NTDs – this is not discussed anywhere else in the review – either in the Results or the Discussion. If this is an interesting and important feature of the evidence, could you discuss it earlier? Are there other aspects of the included studies that would be worth discussing as well?

Reviewer #2: This manuscript presents an important and timely review of assessments of disease surveillance in Africa. It is well written and very comprehensive. The authors conducted a comprehensive systematic review and conformed to PRISMA guidelines. A large number of initial papers were screened, though only 27 were included.

Major revision: As a consequence of the authors' comprehensiveness, the manuscript tends to be wordy and overly narrative, rather than providing a concise analysis of the included studies and themes. I found myself wanting a concise table of themes, countries, etc. Instead, even the tables are overwhelming in content, and more of appendix tables than main text tables. Because of all of this, I think the reader would be aided by adding a "Main Finding" or "Key Themes" or something table/box. In the results section, I struggled with the narrative text. Readers would be much better served by concise reporting of thematic findings with some quantification. There was almost no quantification throughout. Results such as "Ten of the 27 studies indicated challenges with lack of training...".

Minor revisions: Please remove all of the uncommon abbreviations (HW, HP, HF,...).

6. PLOS authors have the option to publish the peer review history of their article (what does this mean?). If published, this will include your full peer review and any attached files.

Reviewer #1: **Yes: **Miranda Cumpston

Reviewer #2: No

---

## [Author Response · Author response to Decision Letter 0]

5 Aug 2020

Reviewer #1: 

Thank you to the authors for producing this detailed and interesting review of recommendations for improving the implementation of disease surveillance in Africa, with a particular focus on system and organisational issues from the perspective of health workers. I am sure this review will be of interest and useful to those working in disease surveillance internationally, identifying common challenges in implementing these systems and recommendations for their improvement, while providing a collated set of evaluation case studies.

I found the review relatively straightforward and clear, although I do have some suggestions to improve the clarity of the methods. Please note that my expertise is in systematic review methodology, and not in disease surveillance systems, so some of my comments may arise from my lack of knowledge of common concepts and terminology in this field.

Author response: Thank you for the remarks.

Introduction, last para (p4, line 93 onward):

• It’s not clear to me what the difference is between the focus of the previous reviews, and the focus of this one. Perhaps this could be more specifically described? If the difference is the healthcare work perspective, perhaps this could be stated more simply. To me, the following descriptions sound the same:

o “critically assessing fundamental recommendations arising from assessment studies regarding IDSR system improvement”

o “key recommendations resulting from IDSR system core, support and attribute functions assessment studies”

Author response (p4, line 93-95): This has been revised to state the focus of the study more clearly. Our review focused on assessing the recommendations to improve performance of the IDSR system core, support and attribute functions based on health workers’ perspectives post-adoption of the revised IDSR guidelines by African countries. 

Table 1 (p5, line 110):

• It might be helpful to write this table using plain language sentences, rather than search terms, to help people understand quickly and easily what you were including. Especially as these terms aren’t necessarily those used in your final search, as included in your supplementary files.

Author response (p5, line 109-114): This has been rectified to briefly summarise what the review included as suggested. 

Eligibility criteria (p7, line 135)

• These criteria could be written more clearly. Make sure it’s clear whether you’re including:

o Only assessments of performance of the IDSR or assessments of any aspect of disease surveillance

o Health workers’ views on the implementation & operation of the system (i.e. studies that explicitly ask health workers to express their views e.g. through surveys or interviews), descriptive characteristics of the implementation and operation of the system that are relevant to health workers (e.g. that could be observed by researchers or policy staff rather than health workers), or any other outcomes (e.g. stats on the effectiveness of surveillance systems or disease outcomes)

Author response (p7, line 143-145): The eligibility criteria has been revised according to the suggestions provided. 

• I’m not sure what the difference is between the studies that meet the main eligibility criteria, and the additional studies that " studies involving record reviews or observations to assess disease surveillance and response systems”. How are these studies different from your included studies, and how are their results treated differently in the review? If they’re not treated differently, then they can be included in the main eligibility criteria. I think you explain this more clearly in your description of include studies in the Results section.

Author response (p7, line 145): The eligibility criteria has been revised. As rightly suggested the studies involving record reviews or observations were not treated differently during analysis and have been included in the main eligibility criteria. 

Search

• The search terms included in the text are not the same as those used in the actual search strategy included in the supplementary documents. Could this be corrected?

Author response (p6, line 132-135): This has been rectified accordingly and the search terms included in the main manuscript have been matched to the sample search strategy included in the supplementary information (refer to Appendix S1. PubMed search strategy).

• If I were drafting this search, I might expect to use some terms such as specific country names (as not all abstracts will specifically refer to Africa), as well as terms relating to evaluation or assessment. I might also consider searching the African Index Medicus. However, I know you have developed this search in collaboration with an information specialist, so you might have made an expert judgement that these would not improve the search.

Author response: The search strategy was revised as suggested and included the search terms “evaluation” and “assessment” across the databases and the overall result of the search increased by 76 additional studies (refer to Appendix S1. PubMed search strategy & Fig 1. Flow chart summarising the systematic review process). Therefore, after screening the additional studies, the final number of reviewed studies increased by three additional studies. On the other hand, the search terms “Africa” and “Sub-Saharan Africa” were preferred to find abstracts of previous assessment studies conducted in either one of the 47 WHO African region countries currently implementing the IDSR system. We initially included specific country names in the search strategy but this hardly improved the search results. In addition, we further conducted a search using African Index Medicus (a developing country health database of the WHO Regional Office for Africa) as suggested and the results matched the relevant studies already retrieved using WHOLIS (World Health Organization Library & Information Networks For Knowledge Database) and PubMed. 

Data extraction (p.7 Line 146)

• The PRISMA statement checklist is a very brief checklist of the presence or absence of methods in a systematic review, and cannot be used for data collection from primary studies. This text should be removed, and a description added of what kinds of data were collected. For example, in your PROSPERO record, you note that “The data to be extracted will be as follows: author's name, article publication year, adoption year of IDSR revised guidelines, target disease/s, study assessment methodology, surveillance functions assessed, key findings and recommendations.” Ideally, a copy of your data collection form could be included as a supplementary document, if available.

Author response (p7, line 150-155): The PRISMA statement checklist has been revised as suggested and a copy of the data extraction form has been included as supplementary information (refer to Appendix S5. Data extraction form).

• Later in the review, in the discussion (p.52, line 478), you discuss developing a matrix of subthemes based on a previous review. I would love to see this discussed here, in the methods section, to describe how you identified themes in the included studies and categorised them according to themes. You could include your matrix as a table or figure. Also, did you add any new items to the matrix that you found in your included studies?

Author response (p7, line 159-164): The matrix that guided thematic synthesis of recommendations derived from the reviewed studies has been provided (refer to Appendix S3. Surveillance system functions). However, apart from the main themes that were based on surveillance core, support and attribute functions, no further items were added to the matrix. 

• It would be helpful to be more specific about exactly what kinds of outcomes you were looking for – that is, outcomes relating to the implementation and operation of the IDSR, but not measures of actual disease surveillance or disease outcomes in the population?

Author response: The anticipated outcomes of the review were to be derived from assessing study recommendations to improve implementation of the IDSR system functions in African countries based on health workers’ views. However, there were no actual disease surveillance measures or population-based disease outcomes assessed in the review. 

• It would be very helpful to provide some additional detail on how you’re identifying which are the key problems identified in each study, and which are the key recommendations. Did you collect every identified problem and every recommendation identified that matched your matrix? Did you select those that you thought were the most important? Did you only select those that matched your original matrix? This is very important so your readers can see how you made your choices and helps them trust your findings.

Author response (p8, line 165-169): Yes, we objectively assessed every finding in the reviewed studies relating to assessment of a surveillance function (i.e. core, support or attribute) matching the predefined themes provided in the matrix, in addition to deriving the corresponding recommendations for improving the surveillance functions. 

• The reason I’m asking for more detail is not because I think the review was not done appropriately, but just to increase transparency and replicability – e.g. if I were a new research assistant, could I follow the instructions in this methods section and successfully replicate the review?

Author response: We agree with the reviewer that the procedure for identifying key recommendations from the reviewed studies based on the matrix of pre-defined themes was not described sufficiently. 

Quality appraisal

• The Nursing Research Appraisal tool used to appraise the included studies is not a true risk of bias assessment tool, and it includes elements that do not relate to study bias, e.g. clarity of writing. However, in the context of this review, which aims to map diverse recommendations, rather than identify a correct effect estimates, I think a more general tool is probably fine.

Author response: The Dearholt and Dang’s John Hopkins Nursing Evidence Appraisal tool, as the reviewer indicated, is a more generalized assessment tool that has been widely used to assess risk of bias in studies included in systematic reviews involving narrative synthesis. This tool was deemed appropriate by the authors in assessing the quality of the reviewed studies with a focus on the assessment methods adopted. However, the authors did not intend to derive quantitative effect estimates from the reviewed studies given it was a qualitative synthesis of diverse study recommendations. 

Results

• Could you include a summary of the study designs found (e.g. cross-sectional, qualitative)

Author response (p10, line 217-222): A brief summary has been included in the result section and further details on the study designs are provided in the assessment methodology column (refer to Table1. Literature summary and quality appraisal). 

• Were there any subthemes identified in your matrix for which no data were found? i.e. that you thought might be issues of concern, but weren’t?

Author response (p7, line 164-166): The subthemes were not pre-defined but rather emerged from the recommendations to improve surveillance functions as described in the reviewed studies. 

• Excluded studies – it would add to the transparency of the review if you could provide a list of excluded studies, with their reasons for exclusion, in addition to the PRISMA flow diagram. This could be, for example, either a full list of studies screened in full text, or a subset of these that readers might be most likely to think

Author response: A summary of the excluded studies has been provided and categorised based on the reasons for exclusion (refer to Appendix S2. Excluded studies). 

Table 4 (p48)

• This table would be easier to read if it were a frequency histogram or bar chart, showing a bar indicating the number of studies with that subtheme, rather than listing all the studies individually. It might even fit on a single page.

Author response (p10, line 231): A bar chart providing a summary of the subthemes and the corresponding number of studies has been provided (refer to Fig 2. Summary of themes, sub-themes and the number of reviewed studies). 

• Should this table be linked from the last para on p.46 (line 455), instead of Table 3? I would recommend it is linked from the Results section, and not first introduced in the Discussion section.

Author response: This has been revised accordingly. A summary of sub-themes with frequency and effect sizes has been provided as supplementary information (refer to Appendix S4. Summary of sub-themes with frequency and intensity effect sizes). 

Discussion

• Should this reference to Table 4 actually be a reference to Table 3? (p47, line 465)

Author response: Thank you for pointing this out. This has been rectified.

• The summary of the results in the Discussion could be more brief – it’s currently almost as long as the Results section (9 pages compared to 10). If there are details here that are important to interpreting the results, could they be included in the Results section? This would then allow the Discussion to be more of a brief overview of main findings, rather than repeating so much detail that might overwhelm readers. This would also make space to include some more discussion of your findings overall.

Author response (p50-58, line 508-681): The discussion section has been revised accordingly by discussing the key findings to make it brief and concise.

• The Discussion could also include some mention of the strengths and limitations of the review design (e.g. those mentioned in the Risk of bias across studies section).

Author response (p57-58, line 673-681): A discussion of the study limitations has been provided.

• Will some recommendations be easier/harder to implement (e.g. some requiring significant additional resources which may not be available, others involving organisational change that’s more achievable?)

Author response (p58-59, line 701-705): Thank you for this suggestion. We have explained this aspect of implementation of the recommendations in the manuscript.

• Who is this data for? Who can or should act on it?

Author response (p59, line 707-710): This has been described in the conclusion section of the manuscript. 

Conclusion

• You noted in the Conclusions that you felt there was not much evidence in NTDs – this is not discussed anywhere else in the review – either in the Results or the Discussion. If this is an interesting and important feature of the evidence, could you discuss it earlier? Are there other aspects of the included studies that would be worth discussing as well?

Author response (p9-10, line 210-213; p56, line 634-641): We observed that diseases classified by WHO as neglected tropical diseases (NTDs) were hardly mentioned as “case diseases” in the reviewed surveillance assessment studies compared to other notifiable conditions. This outcome informed the need for further surveillance assessments considering these conditions as target diseases. Focus was especially on NTDs since they are listed by most African countries as diseases targeted for elimination or of public health importance within the integrated disease surveillance and response systems. 

Reviewer #2: 

This manuscript presents an important and timely review of assessments of disease surveillance in Africa. It is well written and very comprehensive. The authors conducted a comprehensive systematic review and conformed to PRISMA guidelines. A large number of initial papers were screened, though only 27 were included.

Author response: Thank you for the remarks. 

Major revision: As a consequence of the authors' comprehensiveness, the manuscript tends to be wordy and overly narrative, rather than providing a concise analysis of the included studies and themes. I found myself wanting a concise table of themes, countries, etc. Instead, even the tables are overwhelming in content, and more of appendix tables than main text tables. Because of all of this, I think the reader would be aided by adding a "Main Finding" or "Key Themes" or something table/box. 

Author response: In line with the suggestion by the reviewer, a table with the key themes and sub-themes has been included to summarise the findings (refer to Appendix S3. Surveillance system functions & Fig 2. Summary of themes, sub-themes and the number of reviewed studies). Table contents have been revised (refer to Table 1. Literature summary and quality appraisal) and the table summarising sub-themes frequency and intensity effect sizes has been moved from the main manuscript and included as supplementary information (refer to Appendix S4. Summary of sub-themes with frequency and intensity effect sizes).

In the results section, I struggled with the narrative text. Readers would be much better served by concise reporting of 

thematic findings with some quantification. There was almost no quantification throughout. Results such as "Ten of the 27 studies indicated challenges with lack of training...".

Author response (p39-50, line 247-508): Thank you for this suggestion. The results section has been revised to quantify the findings as suggested.

Minor revisions: Please remove all of the uncommon abbreviations (HW, HP, HF,...).

Author response: This has been revised accordingly.

---

## [Decision Letter · Decision Letter 1]

14 Dec 2020

PONE-D-19-27712R1

A systematic review on improving implementation of the revitalised integrated disease surveillance and response system in the African region: a health workers' perspective

PLOS ONE

Dear Dr. Ngetich,

Thank you for submitting your manuscript to PLOS ONE. After careful consideration, we feel that it has merit but does not fully meet PLOS ONE’s publication criteria as it currently stands. Therefore, we invite you to submit a revised version of the manuscript that addresses the points raised during the review process.

We look forward to receiving your revised manuscript.

Kind regards,

Hong-Liang Zhang, M.D., Ph.D.

Academic Editor

PLOS ONE

Reviewers' comments:

Reviewer's Responses to Questions

**Comments to the Author**

1. If the authors have adequately addressed your comments raised in a previous round of review and you feel that this manuscript is now acceptable for publication, you may indicate that here to bypass the “Comments to the Author” section, enter your conflict of interest statement in the “Confidential to Editor” section, and submit your "Accept" recommendation.

Reviewer #1: (No Response)

Reviewer #3: (No Response)

2. Is the manuscript technically sound, and do the data support the conclusions?

Reviewer #1: Yes

Reviewer #3: Partly

3. Has the statistical analysis been performed appropriately and rigorously? 

Reviewer #1: Yes

Reviewer #3: N/A

4. Have the authors made all data underlying the findings in their manuscript fully available?

Reviewer #1: Yes

Reviewer #3: Yes

5. Is the manuscript presented in an intelligible fashion and written in standard English?

Reviewer #1: Yes

Reviewer #3: No

6. Review Comments to the Author

Reviewer #1: All comments have been addressed. My one remaining suggestion is that direct links to the Appendices be included in the relevant sections of the text, to assist readers in finding them.

Reviewer #3: This manuscript emphasized the current disease surveillance in Africa based on the revised IDSR and the methodologies to optimize the system and organizational issues for caregivers and health workers. Authors concluded the implementation of revised IDSR including reporting, feedback, training, supervision, timeliness, and completeness in African regions should be strengthened via optimized strategies and may need further policy reinforcement.

The manuscript has been critically revised in the previous revision. I only have a few suggestions for this Ms as following.

1.The tables are still lengthy for readers. I would suggest the authors further simplify the study aims and key findings for the included researches in Table 1. Meanwhile, I would suggest deleting the Article title in column Table 2, which is redundant with the citations.

2.Reference format should be carefully checked by authors. The current reference list is not consistent.

3.It’s difficult to read the words in Figures. Authors may want to increase the font and clarity.

4.Authors also need to cite the supplementary tables and figures at the appropriate locations in the manuscript.

5.I would authors concise the discussion and conclusion part, the current version is still too wordy.

7. PLOS authors have the option to publish the peer review history of their article (what does this mean?). If published, this will include your full peer review and any attached files.

Reviewer #1: **Yes: **Miranda Cumpston

Reviewer #3: No

---

## [Author Response · Author response to Decision Letter 1]

19 Dec 2020

Thank you for giving us the opportunity to re-submit a revised version of the manuscript titled “A systematic review on improving implementation of the revitalised integrated disease surveillance and response system in the African region: a health workers' perspective” for publication consideration in PLOS One journal. We appreciate the time and effort dedicated by the reviewers to provide insightful feedback on the revised manuscript. We have revised the manuscript considering reviewers’ comments and the provided suggestions. Please find below a point-by-point response to the reviewers’ comments. All page and line numbers cited herein refer to the revised manuscript (without track changes). 

Reviewer #1: 

All comments have been addressed. My one remaining suggestion is that direct links to the Appendices be included in the relevant sections of the text, to assist readers in finding them.

Author response (p6, line 123 & 135; p7, line 148, 152, 155 & 160; p8, line 173; p41, line 514): The manuscript has been revised as suggested by providing relevant links to all supplementary files and tables in the relevant sections within the text. 

Reviewer #2: 

This manuscript emphasized the current disease surveillance in Africa based on the revised IDSR and the methodologies to optimize the system and organizational issues for caregivers and health workers. Authors concluded the implementation of revised IDSR including reporting, feedback, training, supervision, timeliness, and completeness in African regions should be strengthened via optimized strategies and may need further policy reinforcement.

The manuscript has been critically revised in the previous revision. I only have a few suggestions for this Ms as following.

1. The tables are still lengthy for readers. I would suggest the authors further simplify the study aims and key findings for the included researches in Table 1. Meanwhile, I would suggest deleting the Article title in column Table 2, which is redundant with the citations.

Author response (p12, line 241 & p21, line 251): Table 1 has been revised by providing a brief summary of the study aims and key findings retrieved from the reviewed studies. In addition, the column “Article title” in Table 2 has been deleted as suggested. 

2. Reference format should be carefully checked by authors. The current reference list is not consistent.

Author response (p49, line 685): All references have been updated accordingly to ensure consistency. 

3. It’s difficult to read the words in Figures. Authors may want to increase the font and clarity.

Author response (p9, line 202 & p10, line 233): The font size for labels and words contained in all figures have been increased to ensure clarity. 

4. Authors also need to cite the supplementary tables and figures at the appropriate locations in the manuscript.

Author response (p6, line 123 & 135; p7, line 148, 152, 155 & 160; p8, line 173; p41, line 514): The manuscript has been revised as suggested by citing all supplementary files and tables in the appropriate sections within the text. 

5. I would authors concise the discussion and conclusion part, the current version is still too wordy.

Author response (p41-47, line 515-665): The discussion section has been revised accordingly by only discussing the key findings. Similarly, the manuscript conclusion has been revised as suggested.

---

## [Decision Letter · Decision Letter 2]

23 Feb 2021

PONE-D-19-27712R2

A systematic review on improving implementation of the revitalised integrated disease surveillance and response system in the African region: a health workers' perspective

PLOS ONE

Dear Dr. Ngetich,

Thank you for submitting your manuscript to PLOS ONE. After careful consideration, we feel that it has merit but does not fully meet PLOS ONE’s publication criteria as it currently stands. Therefore, we invite you to submit a revised version of the manuscript that addresses the points raised during the review process.

We look forward to receiving your revised manuscript.

Kind regards,

Hong-Liang Zhang, M.D., Ph.D.

Academic Editor

PLOS ONE

Journal Requirements:

Reviewers' comments:

Reviewer's Responses to Questions

**Comments to the Author**

1. If the authors have adequately addressed your comments raised in a previous round of review and you feel that this manuscript is now acceptable for publication, you may indicate that here to bypass the “Comments to the Author” section, enter your conflict of interest statement in the “Confidential to Editor” section, and submit your "Accept" recommendation.

Reviewer #3: All comments have been addressed

Reviewer #4: (No Response)

2. Is the manuscript technically sound, and do the data support the conclusions?

Reviewer #3: Yes

Reviewer #4: Yes

3. Has the statistical analysis been performed appropriately and rigorously? 

Reviewer #3: Yes

Reviewer #4: Yes

4. Have the authors made all data underlying the findings in their manuscript fully available?

Reviewer #3: Yes

Reviewer #4: Yes

5. Is the manuscript presented in an intelligible fashion and written in standard English?

Reviewer #3: Yes

Reviewer #4: No

6. Review Comments to the Author

Reviewer #3: (No Response)

Reviewer #4: According to previous reviewers' comments and authors' answers, the manuscript has been critically revised. However, authors should further concise the Table 1 and the conclusion part. In the next round of review, authors should submit a clear vision of manuscript without changing traces.

7. PLOS authors have the option to publish the peer review history of their article (what does this mean?). If published, this will include your full peer review and any attached files.

Reviewer #3: No

Reviewer #4: No

---

## [Author Response · Author response to Decision Letter 2]

25 Feb 2021

Response to academic editor and reviewer comments

Manuscript PONE-D-19-27712R2

Title: A systematic review on improving implementation of the revitalised integrated disease surveillance and response system in the African region: a health workers' perspective”

Authors:

Arthur Kipkemoi Saitabau Ng'etich (arthursaitabau@yahoo.com)

Kuku Voyi (kuku.voyi@up.ac.za) 

Ruth Kirinyet (kirinyet.ruth@ku.ac.ke) 

Clifford Maina Mutero (cmutero@icipe.org) 

Dear Editor,

Thank you for giving us the opportunity to re-submit a revised version of our manuscript for publication consideration in PLOS One journal. We appreciate the time and effort dedicated by the academic editor and reviewers to provide insightful feedback on the revised manuscript. We have revised the manuscript considering all the comments provided. Please find below a point-by-point response to the comments. All page and line numbers cited herein refer to the revised manuscript (without track changes). 

Academic Editor: 

Journal Requirements:

Author response: The reference list has been reviewed to ensure all references are complete and correct. However, two references (refer below) in the previous manuscript were replaced with a relevant and updated reference (References, line 879-881, page 54). We further checked all references and can confirm to the best of our knowledge that none of the articles cited were retracted.

-Franco LM, Fields R, Mmbuji PK, Posner S, Mboera LE. Situation analysis of infectious disease surveillance in two districts in Tanzania 2002. Working Paper; 2003.

-Pond B, El Sakka H, Wamala J, Lukwago L. Mid-term evaluation of the integrated disease surveillance and response project. Washington, DC: USAID. 2011.

Reviewers’ comments: 

1. If the authors have adequately addressed your comments raised in a previous round of review and you feel that this manuscript is now acceptable for publication, you may indicate that here to bypass the “Comments to the Author” section, enter your conflict of interest statement in the “Confidential to Editor” section, and submit your "Accept" recommendation.

Reviewer #3: All comments have been addressed

Reviewer #4: (No Response)

Author: No Response 

2. Is the manuscript technically sound, and do the data support the conclusions?

Reviewer #3: Yes

Reviewer #4: Yes

Author: No Response 

3. Has the statistical analysis been performed appropriately and rigorously? 

Reviewer #3: Yes

Reviewer #4: Yes

Author: No Response 

4. Have the authors made all data underlying the findings in their manuscript fully available?

Reviewer #3: Yes

Reviewer #4: Yes

Author: No Response 

5. Is the manuscript presented in an intelligible fashion and written in standard English?

Reviewer #3: Yes

Reviewer #4: No

Author response: The manuscript has been be revised accordingly by checking and correcting all typographical and grammatical errors.

6. Review Comments to the Author

Reviewer #3: (No Response)

Reviewer #4: According to previous reviewers' comments and authors' answers, the manuscript has been critically revised. However, authors should further concise the Table 1 and the conclusion part. In the next round of review, authors should submit a clear vision of manuscript without changing traces.

Author response: Table 1 and the conclusion section have been revised as suggested by the reviewer. The final version of the revised manuscript is without any track changes. 

All authors reviewed the final version of the revised manuscript and ensured it adheres to all editorial requirements. We hope that our response to the comments provided render the revised manuscript suitable for publication in PLOS One journal.

Yours Sincerely,

Arthur Kipkemoi Saitabau Ng’etich

(Corresponding author)

---

## [Editor Report · Decision Letter 3]

10 Mar 2021

A systematic review on improving implementation of the revitalised integrated disease surveillance and response system in the African region: a health workers' perspective

PONE-D-19-27712R3

Dear Dr. Ngetich,

We’re pleased to inform you that your manuscript has been judged scientifically suitable for publication and will be formally accepted for publication once it meets all outstanding technical requirements.

Kind regards,

Hong-Liang Zhang, M.D., Ph.D.

Academic Editor

PLOS ONE
---

## [Editor Report · Acceptance letter]

11 Mar 2021

PONE-D-19-27712R3 

A systematic review on improving implementation of the revitalised integrated disease surveillance and response system in the African region: a health workers' perspective 

Dear Dr. Ngetich:

I'm pleased to inform you that your manuscript has been deemed suitable for publication in PLOS ONE. Congratulations! Your manuscript is now with our production department. 

Kind regards, 

on behalf of

Dr. Hong-Liang Zhang 

Academic Editor

PLOS ONE